# *OsMADS1* Regulates Grain Quality, Gene Expressions, and Regulatory Networks of Starch and Storage Protein Metabolisms in Rice

**DOI:** 10.3390/ijms24098017

**Published:** 2023-04-28

**Authors:** Zhijian Liu, Penghui Li, Lan Yu, Yongzhi Hu, Anping Du, Xingyue Fu, Cuili Wu, Dagang Luo, Binhua Hu, Hui Dong, Haibo Jiang, Xinrong Ma, Weizao Huang, Xiaocheng Yang, Shengbin Tu, Hui Li

**Affiliations:** 1Chengdu Institute of Biology, Chinese Academy of Sciences, Chengdu 610041, China; 2University of Chinese Academy of Sciences, Beijing 100049, China; 3College of Ecology and Environment, Chengdu University of Technology, Chengdu 610059, China; 4Crop Research Institute, Sichuan Academy of Agricultural Sciences, Chengdu 610066, China; 5Biotechnology and Nuclear Technology Research Institute, Sichuan Academy of Agricultural Sciences, Chengdu 610066, China; 6State Key Laboratory of Crop Genetics and Germplasm Enhancement, Nanjing Agricultural University, Nanjing 210095, China

**Keywords:** *OsMADS1*, grain quality, starch metabolism, seed storage protein metabolism, rice

## Abstract

*OsMADS1* plays a vital role in regulating floret development and grain shape, but whether it regulates rice grain quality still remains largely unknown. Therefore, we used comprehensive molecular genetics, plant biotechnology, and functional omics approaches, including phenotyping, mapping-by-sequencing, target gene seed-specific RNAi, transgenic experiments, and transcriptomic profiling to answer this biological and molecular question. Here, we report the characterization of the ‘Oat-like rice’ mutant, with poor grain quality, including chalky endosperms, abnormal morphology and loose arrangement of starch granules, and lower starch content but higher protein content in grains. The poor grain quality of Oat-like rice was found to be caused by the mutated *OsMADS1*^Olr^ allele through mapping-by-sequencing analysis and transgenic experiments. OsMADS1 protein is highly expressed in florets and developing seeds. Both OsMADS1-eGFP and OsMADS1^Olr^-eGFP fusion proteins are localized in the nucleus. Moreover, seed-specific RNAi of *OsMADS1* also caused decreased grain quality in transgenic lines, such as the Oat-like rice. Further transcriptomic profiling between Oat-like rice and Nipponbare grains revealed that *OsMADS1* regulates gene expressions and regulatory networks of starch and storage protein metabolisms in rice grains, hereafter regulating rice quality. In conclusion, our results not only reveal the crucial role and preliminary mechanism of *OsMADS1* in regulating rice grain quality but also highlight the application potentials of *OsMADS1* and the target gene seed-specific RNAi system in improving rice grain quality by molecular breeding.

## 1. Introduction

Rice (*Oryza sativa* L.) is one of the top three staple crops worldwide, which feeds over half of the global population. Like yield, grain quality is one of the most concerning key agronomic traits for rice consumers, sellers, processors, producers, and breeders, which largely determines the taste, nutritional value, and price of rice. The rice grain quality is mainly composed of milling quality, appearance quality, cooking and sensory quality, and nutrition quality [1], which is comprehensively controlled by a series of key regulatory genes and environmental factors. Thus, mapping and cloning key genes regulating grain quality from important rice germplasm resources, studying and uncovering the corresponding regulatory mechanisms, and subsequent application of these genes in breeding for rice grain quality are indispensable and helpful for scientists and breeders to develop superior-quality rice varieties.

Some transcription factors, such as MADS-box family members known to regulate inflorescence and spikelet development in rice, are also thought to be related to grain quality regulation, but the detailed underlying regulatory mechanisms are still to be explored. It is interesting that some MADS-box genes such as *OsMADS1*, *OsMADS6*, and *OsMADS7* are specifically and highly expressed both in panicles and seeds [2], suggesting these genes could also be related to the regulation of seed development and grain quality formation. This deduction was partly confirmed by the abnormal phenotypes in *OsMADS6* mutant with both abnormal floral organs and shriveled seeds in which the grain quality, starch filling, and expressions of ADP-glucose pyrophosphorylase genes were decreased, but the seed protein content was increased [3]. In addition, another MADS-box gene, *OsMADS29,* was found to be specifically expressed in seeds, including nucellus, dorsal-vascular trace, embryo, and endosperm. Os*MADS29*-RNAi (Os*MADS29*-RNA interference) plants exhibited suppressed starch biosynthesis and reduced grain-filling rate, loosely packed starch granules, and shrunken grains. Expression of *OsMADS29* is induced by auxin, and OsMADS29 transcription factor regulates degradation of maternal tissues, including ovule, by directly binding to the downstream programmed cell death genes in these maternal tissues, thereby controlling subsequent seed development and affecting the grain quality formation [4]. In addition, Zhang et al. [5] reported that high temperature at the early filling stage greatly induced the expression of *OsMADS7* and suppression of it in rice endosperm stabilized amylose content, possibly by maintaining a relatively low filling rate and high expression of the encoding gene of GBSS I (granule-bound starch synthase I) under high-temperature stress.

Furthermore, extensive studies have revealed that another class E floral homeotic gene of the MADS-box family, *OsMADS1*, plays important roles in regulating floral organ development in rice by regulating the floral meristem specification and floral organ identity [6,7,8,9]. Several recent studies further found out that OsMADS1 also controls grain shape by interactions with G-protein (guanine nucleotide-binding protein) subunits, including DEP1 (dense and erect panicle 1) and GS3 (grain size 3) to collaboratively regulate expressions of underlying downstream target genes related to grain shape and development [10,11,12,13].

After the first *OsMADS1* mutant, *lhs1* (*leafy hull sterile1*) was reported by Kinoshita et al. [14], at least 11 mutants and four NILs (Near-isogenic lines) of *OsMADS1* including *lhs1* [7,14], *nsr* (*naked seed rice*) [8], *NF1019*, *ND2920*, *NE3043* and *NG778* [6], *osmads1-z* [9], *afo* (*abnormal floral organs*) [15], *ohms1* (*open hull and male sterile 1*) [16], *cy15* [17] and Olr (Oat-like rice) [10], NIL (SLG) [13], WYJ7 (Wuyungeng No. 7)*-lgy3-dep1–1*, RD23*-lgy3-gs3* and PA64S (Peiai 64S)/9311 (Yangdao No. 6)*-lgy3-gs3* [11] have been reported so far. Thereinto, Olr is a spontaneous and severe *OsMADS1* mutant, which was named for its unique grain shape, which highly resembles oat grains. Olr displayed abnormal floral organs, open hulls formed by remarkably elongated leafy lemmas and paleae, occasionally formed conjugated twin brown rice, aberrant grain shape, low seed setting rate, slow grain-filling rate, low 1000-brown rice weight, and extremely low yield [10].

However, it’s interesting that the grain quality was only analyzed in NIL (SLG) [13], WYJ7*-lgy3-dep1–1*, RD23*-lgy3-gs3*, PA64S/9311*-lgy3-gs3* [11], which may be partly due to unavailable seeds caused by sterility or extremely low fertility of these mutants. These four rice NILs carry the same mutated allele of *OsMADS1*, *OsMADS1^lgy3^/OsLG3b^SLG^*, but the effect of this allele on the grain quality of these NILs was divided into two distinct categories. There is no significant difference in chalkiness between grains of NIL (SLG) and its receptor parent Nipponbare, which indicates that *OsLG3b^SLG^* did not affect grain quality in Nipponbare [13]. However, WYJ7*-lgy3-dep1–1*, RD23*-lgy3-gs3,* and PA64S/9311*-lgy3-gs3* grains all displayed lower chalkiness compared with their respective receptor parent, WYJ7, RD23, and LYPJ (Liangyoupeijiu, PA64S/9311), which indicates that *OsMADS1^lgy3^* affected and improved grain quality in WYJ7, RD23, and LYPJ [11]. Interestingly, the question of whether *OsMADS1* regulates grain quality is still contradictory and obscure.

Thus, it is necessary to analyze the grain quality of more mutants or accessions carrying different mild, moderate, or severe *OsMADS1* alleles to confirm whether *OsMADS1* regulates grain quality and further uncover the regulatory mechanism. In this study, we analyzed the grain quality of the *OsMADS1*^Olr^ mutant (Olr) and identified a correlation between the mutation in the *OsMADS1* gene and the chalkiness phenotype. The Olr mutant displays poor grain quality with chalky endosperms, abnormal morphology, and loose arrangement of starch granules, with lower starch content but higher protein content in grains. Moreover, *OsMADS1* has a crucial role in regulating rice quality by coordinating gene expressions and regulatory networks of starch and storage protein metabolisms in rice grains. Therefore, this study also highlighted the application potentials of *OsMADS1* and the target gene seed-specific-RNAi system in rice breeding to develop superior-quality rice varieties.

## 2. Results

### 2.1. Olr Exhibits Pleiotropic and Heritable Phenotypes of Chalky Endosperms and Abnormal Grain Shape

Our previous study revealed that Olr exhibits abnormal grain shape and long grains determined by elongated open lemmas and paleae, which is accompanied by lower 1000-brown rice weight and slower grain-filling rate [10]. To explore if the appearance quality of Olr grains was also affected, we initially analyzed the appearance quality of endosperms between Olr and the control variety, Nip (Nipponbare), which exhibits normal grain quality as well as grain shape and is a wildly used model variety in rice. Compared with Nip, endosperms of Olr are not only apparently smaller but also exhibit more grain chalkiness with opaque white-core in the inner endosperm, as indicated by an extremely significant and higher percentage of endosperms with chalkiness and degree of endosperm chalkiness (Figure 1A–C). Furthermore, the Olr displayed stable and heritable Olr phenotypes with chalky endosperms and abnormal grain shape in every generation in Sichuan and Hainan provinces in China by field observation, indicating the pleiotropic phenotypes are heritable and controlled by genetic factors.

### 2.2. The Chalky Endosperms of Olr Are Caused by Abnormal Morphology and Loose Arrangement of Starch Granules

Further cross-section analysis of Olr and Nip brown rice by dissecting microscope revealed that although both of their outer endosperms are translucent and similar to each other, the inner endosperm of Olr displayed a chalky white-core (Figure 1D,H). Scanning electron microscope analysis showed that the chalky inner area in Olr endosperm has relatively lower electron density than the translucent Nip endosperm and translucent outer area of Olr endosperm, which suggests that the shape and arrangement of starch granules in the chalky inner area of Olr endosperm may be affected (Figure 1E,I). A magnified view reveals that in contrast to the tightly arranged larger polyhedral starch granules of translucent endosperm in Nip, some small polyhedral and spherical starch granules arranged loosely with large air spaces were observed in the chalky endosperm of Olr (Figure 1F,G,J,K). It suggests that not only the morphology and arrangement of starch granules but also the formation of starch granules were affected in the chalky endosperm of Olr. Therefore, the occurrence of chalkiness in Olr endosperms was caused by the abortive or compromised formation of starch granules, resulting in abnormal morphology and loose arrangement of starch granules.

### 2.3. The Grain Quality of Olr with Relatively Lower Starch Content but Higher Protein Content Is Linked with Its Grain Shape

Subsequently, the starch and storage protein contents of brown rice from Olr, Nip, and their progeny plants were determined. The percentages of total starch, amylose, and amylopectin contents of Olr brown rice are 86.66 ± 0.74%, 20.43 ± 0.15%, and 66.23 ± 0.81%, which are lowered by 10.51%, 4.55%, and 12.35% than that of the Nip brown rice. By contrast, the Olr brown rice accumulated an extremely high content of storage protein, which is 14.13 ± 0.07%, and increased by 51.77% than that of the Nip brown rice (Table 1). These results that starch and storage protein contents were changed in Olr grains imply that metabolism and accumulation of starch and storage protein may also be changed. It is likely that decreased starch content but increased storage protein content affected normal formation, morphology, and arrangement of starch granules, thereby causing chalk endosperms and poor grain quality of Olr.

To figure out whether changes in grain quality, including starch and storage protein contents in Olr grains, were linked with its abnormal grain shape, a linkage analysis between the two traits was performed. We measured the starch and storage protein contents of brown rice from plans of F_2_ and F_3_ segregating populations derived from a cross combination between Olr and Nip. If the two traits are linked with each other that indicates that they are caused by the same mutated gene in Olr. However, the two traits are caused by different genes. The test results were very similar to the comparison of starch and storage protein contents between Olr and Nip. Both F_2_ and F_3_ plants showing Olr phenotypes with poor grain quality and abnormal grain shape exhibited extremely lower contents of total starch, amylose, and amylopectin but extremely higher content of storage protein as a whole, compared with that of both F_2_ and F_3_ plants showing normal grain quality and grain shape, respectively. For example, compared with the plants showing normal phenotypes in the F_3_ population, the percentage contents of total starch, amylose, and amylopectin in brown rice from plants showing Olr phenotypes are 75.71 ± 1.53%, 15.95 ± 0.04%, and 59.76 ± 1.49%, which is lowered by 14.28%, 11.19%, and 15.07% respectively. However, these F_3_ plants showing Olr phenotypes accumulated an extremely high content of storage protein in the brown rice, which is 14.90 ± 0.03%, and increased by 97.09% compared with that of F_3_ plants showing normal phenotypes (Table 1). These results collectively indicate that the grain quality of Olr, with relatively lower starch content but higher protein content, is linked with its grain shape. We further speculated that both the poor grain quality and abnormal grain shape of Olr were caused by the pleiotropic effects of the same mutated gene, *OsMADS1*^Olr^ [10].

### 2.4. OsMADS1 Has a Direct Role in Controlling Grain Quality

In order to examine whether the poor grain quality of Olr was caused by the mutated *OsMADS1*^Olr^ gene, we tried several times to perform a complementation test by expressing the wild-type *OsMADS1* gene in Olr and finally obtained positive transgenic plants (Figure 2). The grain shape of the eight T_0_ positive plants was partly recovered with a closed hull, shortened grain length, and grain width, which is similar to the normal rice grains of Nip (Figure 2A). In the Nip and T_0_ negative control plant, the 174-bp genomic fragment of the endogenous wild-type *OsMADS1* and mutated *OsMADS1*^Olr^ gene were identified by PCR test. By contrast, both the 174-bp genomic fragment of the endogenous mutated *OsMADS1*^Olr^ gene and the 84-bp cDNA fragment of the exogenous wild-type *OsMADS1* gene (cDNA) harbored by the p*Ubi*::*OsMADS1* vector were identified in these eight T_0_ positive plants (Figure 2B,C). Further statistical analysis showed that grain length, grain width, grain length-to-width ratio, and grain thickness of these eight T_0_ positive plants were significantly decreased than those of the T_0_ negative control plant but more similar to the normal rice grains of Nip as a whole (Figure 2D,E).

These results showed that the grain shape of Olr was rescued by introducing the wild-type *OsMADS1* gene, which confirmed that the abnormal grain shape of Olr was caused by its mutated *OsMADS1*^Olr^ gene. However, we were unable to know whether the poor grain quality of Olr was recovered or not due to spikelet sterility or failure in the germination of occasionally produced several seeds of the transgenic plants. As shown in Figure 2, both control plants and six of the eight T_0_ positive plants (C-28, C-30, C-3, C-4, C-5, and C-10) were sterile; only C-9 and C-6 plants occasionally produced several seeds but failed to germinate (Figure 2A, indicated by white arrows).

When the flowering and grain-filling stages are in the summer seasons, the seed-setting rate of Olr mutant is around 15% in Chengdu city, Sichuan province of China in different years. However, both T_0_ control and positive plants exhibited spikelet sterility or extremely low fertility (0.36% of C-9 and 1.05% of C-6 plants, respectively). So, we speculated that spikelet sterility or extremely low fertility of these T_0_ plants could be possibly attributed to the relatively cold weather in late October, early and middle November in Chengdu during the flowering, grain filling, and seed development stages of these plants (Appendix A).

To study if *OsMADS1* has a direct role in controlling grain quality, we further used an *OsMADS1* seed-specific RNAi system and analyzed the grain appearance quality of descendant lines of p*OsTip3 (tonoplast intrinsic protein 3)*::*OsMADS1*-RNAi plants in details. Compared with wild-type plants (Nip), the three homozygous p*OsTip3*::*OsMADS1*-RNAi T_3_ lines (SRi11, SRi14, and SRi16) showed a slightly darker brown color in the T_4_ brown rice (Figure 3A–D). Furthermore, all these three lines exhibited more chalkiness, indicated by consistently and greatly increased percentage of brown rice with chalkiness, percentage of endosperms with chalkiness, and degree of endosperm chalkiness than that of wild-type plants (Figure 3I–K). Thus, seed-specific RNAi of the *OsMADS1* gene resulted in poor grain appearance quality in the transgenic lines, which is consistent with the poor grain quality of Olr and further indicates that *OsMADS1* has a direct role in regulating grain quality in rice.

### 2.5. Verifying the Causal Gene of Grain Quality Phenotype in Olr through Mapping-by-Sequencing

Liu et al. [11] reported that three NILs of the mutated *OsMADS1^lgy3^* gene, WYJ7*-lgy3-dep1–1*, RD23*-lgy3-gs3* and PA64S/9311*-lgy3-gs3*, all displayed improved grain quality, which is contradictory with the poor grain quality of Olr and *OsMADS1* seed-specific RNAi. To confirm whether the poor grain quality of Olr was indeed caused by the *OsMADS1*^Olr^ gene, we further mapped and isolated the causal gene using the mapping-by-sequencing method based on a constructed F_2_ segregating population by a cross between Olr and Nip. Two pooled DNA libraries from maternal plant Olr (MO-bulk) and parental plants Nip (PN-bulk), and another two groups of their F_2_ progeny plants showing Olr phenotypes (O-bulk) or normal phenotypes (N-bulk) were constructed, and sequenced. Screened SNPs (single nucleotide polymorphisms) between Olr and Nip were used to calculate the SNP-index of O-bulk and N-bulk and the corresponding ∆(SNP-index) between the two bulks. Then, linkage maps between SNPs and the candidate causal gene in the 12 chromosomes of rice were constructed by the SNP-index and ∆(SNP-index) and were subsequently used to identify the location of the candidate causal gene (Section 4.7.).

Both an SNP-index peak of 1.0 and a corresponding SNP-index trough of 0.37 were simultaneously identified at the same location within the same 2.15 interval (5.26–7.41 Mb) of rice chromosome 3 in the SNP-index plots of O-bulk and N-bulk, respectively. The corresponding ∆(SNP-index) peak between O-bulk and N-bulk is 0.63 and located at the same location within the 2.15 interval above the 99% confidence interval. Therefore, the causal candidate gene of Olr grain quality was identified in the 5.26–7.41 Mb interval of rice chromosome 3 (Figure 4A,B). Within the 2.15 Mb candidate region, four candidate causal SNPs distributed in three candidate genes were identified according to a series of stringent screening criteria (Appendix A and Figure 4C).

In the three candidate genes, *OsMADS1* (*Os03g0215400*) was previously reported to regulate flower development and grain shape [7,11]. Further protein expression pattern of OsMADS1 showed that its expressions were negligible in flag leaves but high in both florets and developing seeds. Furthermore, OsMADS1 exhibited a continuously decreasing expression pattern in developing seeds from 1–6 to 12–18 DAF seeds (Figure 4E). This result suggests the possibility that *OsMADS1* is involved in the regulation of seed development and grain quality formation.

In the mutated *OsMADS1*^Olr^ allele of Olr, the 80th SNP mutated from G to A in the first exon, further verified by sequencing, resulting in the amino acid mutation of the 27th amino acid in the MADS-box domain of the encoding OsMADS1 protein from conserved and nonpolar G (glycine) to polar D (aspartic acid) (Figure 4C,D). Subsequent 3D protein structure prediction and comparison showed that the structure of the mutated 27th aspartic acid and its side chain in the MADS-box domain of OsMADS1^Olr^ was altered compared with that of the wild-type OsMADS1 protein (Figure 5A–D). This result suggests that mutation of the aspartic acid in the OsMADS1^Olr^ protein may affect its protein structure, including the MADS-box domain, thereby affecting the binding ability to its target genes and normal regulatory roles to seed development and grain quality formation. In addition, we detected the subcellular localization of OsMADS1-eGFP (OsMADS1-enhanced green fluorescent protein) and OsMADS1^Olr^-eGFP fusion proteins in rice protoplasts and found out that both of them were localized in the nucleus (Figure 5E–P), which is not only consistent with the nucleus localization of MADS-box transcription factors but also indicates that mutation in the MADS-box domain of OsMADS1^Olr^ didn’t affect its nucleus localization. Collectively, these results confirmed that *OsMADS1*^Olr^ is the causal gene of the poor grain quality in Olr.

### 2.6. Overview of mRNA-Seq Results of Developing Grains from Nip and Olr

To explore the gene regulatory networks and mechanism of *OsMADS1* in regulating rice grain quality, we performed a comprehensive transcriptomic analysis in developing grains between Nip and Olr by Illumina high-throughput transcriptome sequencing. Overview of mRNA-seq results indicate that the mRNA-seq data with sufficient and high-quality reads, high mapping ratios of these reads to the Nip reference genome, and good correlation and similarity within biological replicates can be used for the downstream gene expression detection and differentially expressed gene analysis (Appendix A, and Appendix A).

### 2.7. Identification and Analysis of Differentially Expressed Genes (DEGs) between Nip and Olr Grains

DEGs analysis in developing grains between Olr and Nip was subsequently performed to explore the gene regulatory networks and mechanism of *OsMADS1* in regulating rice grain quality. The numbers of DEGs and FPKM (Fragments per kilobase of transcript per million mapped reads) values of the corresponding genes obtained from grain samples at 1 DAF, 6 DAF, 12 DAF, 18 DAF, and 24 DAF between Nip and Olr were listed in Appendix A and Appendix A, respectively. Overall, the numbers of upregulated DEGs are all lower than the numbers of downregulated DEGs in NS01_vs_OS01, NS06_vs_OS06, NS12_vs_OS12, NS18_vs_OS18, and NS24_vs_OS24 comparison groups, with a total of 13,550 upregulated and 20,822 downregulated DEGs of the five comparison groups (Appendix A). We further analyzed the DEGs using a Venn diagram to explore unique, common, and specific DEGs in different comparison groups and disjointed subgroups. In the comparison groups between Nip and Olr, 16294 unique DEGs (7635 upregulated and 9775 downregulated) and 1619 common DEGs (336 upregulated and 1268 downregulated) were identified (Appendix A). Collectively, the results of the Venn diagram (Appendix A) and statistics of DEGs (Appendix A) are consistent with each other, which reflects the different transcriptomic profiles of developing grains between Olr and Nip.

### 2.8. GO and KEGG Enrichment Analysis of DEGs between Nip and Olr Grains

To isolate related GO (Gene Ontology) terms and KEGG (Kyoto Encyclopedia of Genes and Genomes) pathways of *OsMADS1* in regulating rice grain quality, GO and KEGG functional enrichment analysis was employed for DEGs between Nip and Olr samples. In summary, the enriched GO terms and KEGG pathways are mutually corroborated and complementary to each other in general, and some KEGG pathways overlap with part of the GO terms, especially in the GO BP (biological processes) and GO MF (molecular functions) terms. Furthermore, the enriched GO terms and KEGG pathways were concentrated in the functional categories, including carbon fixation and nitrogen assimilation, carbon and nitrogen metabolism, synthesis and transport of carbohydrate and amino acid, synthesis, processing, transport, and accumulation of starch as well as proteins. Moreover, these enriched functional categories are closely involved in the synthesis, metabolism, and accumulation of starch and storage protein in rice grains, which suggests that *OsMADS1* likely controls the starch and storage protein contents as well as grain quality through these enriched functional categories (Appendix A).

### 2.9. Analysis of the Dynamic Gene Expression Patterns between Nip and Olr Grains

We found out that the identified 30 Nip and 36 Olr robust clusters showed diverse temporal expression profiles by self-organizing maps (SOM) analysis (Appendix A). More GO items and KEGG pathways were enriched in the 36 Olr robust clusters than that in the 30 Nip robust clusters by GO and KEGG functional enrichment analysis (Appendix A). Further identified storage protein-related GO items and KEGG pathways were found to be closely involved in the biological processes and metabolic pathways of synthesis, processing, transport, and accumulation of storage protein in grains (Appendix A). Furthermore, identified starch-related GO items and KEGG pathways were mainly related to starch synthesis, metabolism, and regulation. Interestingly, these starch-related GO items, KEGG pathways, and corresponding genes are predominantly enriched in the No. 5 robust cluster of Nip, and the most abundant in the No. 25 robust cluster of Olr, separately (Appendix A). The No. 25 robust cluster in Olr showed overall lower gene expression levels than that of the No. 5 robust cluster in Nip. This result is well consistent with our previous results that Olr exhibited a slower grain-filling rate and lower dry weight accumulation than that of Nip during grain development, which resulted in the remarkably lower 1000-brown rice weight of Olr (Appendix A) [10]. Considering all these results and the fact that starch accounts for up to about 80% of the grain weight of rice, it suggests that the lower starch content, abnormal morphology and loose arrangement of starch granules, lighter 1000-brown rice weight, and poor appearance quality of Olr grains than that of Nip grains is related to the differences of expression patterns of starch related genes between them during grain development.

### 2.10. Analysis of Gene Expressions and Regulatory Networks of Starch Metabolism Pathways between Olr and Nip Grains

We analyzed gene expressions in starch metabolism pathways between Olr and Nip grains and found out that expressions of the 20 starch biosynthesizing genes in Olr were downregulated compared with Nip as a whole (Figure 6 and Appendix A). Thereinto, *OsAGPS1*, *OsAGPS2 (ADP-glucose pyrophosphorylase small subunit 1, 2*), *OsAGPL1,* and *OsAGPL2* (*ADP-glucose pyrophosphorylase large subunit 1, 2)* encode the subunits of ADP-glucose pyrophosphorylase, which is the rate-limiting enzyme of the starch biosynthesis pathway. All these four genes displayed downregulated expressions as a whole. In addition, transcripts of *OsSUT2*, *OsGPT1,* and *OsBT1,* which encode the sucrose transporter 2, glucose-6P translocator 1, and ADP-glucose transporter 1, respectively, were decreased in Olr grains on average. Interestingly, the negative regulatory gene *RSR1* of starch biosynthesis showed obviously upregulated expression, but expressions of the other six positive regulatory genes, including *RSR1* and *OsNF-YB1,* were apparently downregulated in Olr grains as a whole. By contrast, about two-thirds (67.27%) of the gene expression ratio grids in the starch degradation pathway displayed upregulated expression levels in Olr grains (Figure 6A,B). On the whole, the mutation of *OsMADS1* in Olr resulted in an overall decline but an increase in gene expressions in starch biosynthesis and degradation pathways. Thereby, it may reduce the starch biosynthesis but increase the starch degradation, leading to lower starch content and poor grain quality of Olr. Collectively, we concluded that *OsMADS1* could control grain quality by coordinating gene expressions and regulatory networks of starch metabolism pathways in rice grains.

### 2.11. Analysis of Gene Expressions and Regulatory Networks of Storage Protein Biosynthesis and Accumulating Pathways between Olr and Nip Grains

We further analyzed gene expressions in storage protein biosynthesis and accumulating pathways between Olr and Nip grains and found out that except for *GluB2 (glutelin precursor B2)*, all the rest 11 glutelin precursor and one globulin genes exhibited obviously upregulated expressions at 6 DAF and 12 DAF grains in Olr. On the whole, expressions of prolamins genes were upregulated at 6 DAF grains. However, the 11 downstream genes controlling seed storage protein processing, trafficking, and accumulation displayed relatively smaller expression changes. On the whole, expression levels of the 26 seed storage protein biosynthesizing and regulating genes were downregulated at 1 DAF, 18 DAF, and 24 DAF grains but upregulated at 6 DAF and 12 DAF grains, with 84.00% and 73.07% upregulated genes, respectively. Expressions of these genes in 1 DAF grains in both Olr and Nip are relatively very low (Figure 7 and Appendix A). However, seed storage protein accumulation can be detected at 4 DAF grains and increases quickly from 6 DAF to 12 DAF grains and continues to increase slowly from 12 DAF grains until the grains are at a mature stage [18]. Thus, mutation of *OsMADS1* in Olr resulted in upregulation of gene expressions and disruption of regulatory networks of the seed storage protein biosynthesis during its quick accumulation stage, which may result in increased seed storage protein content and poor grain quality of Olr grains. Taking all these results together, we concluded that *OsMADS1* could control grain quality by coordinating gene expressions and regulatory networks of starch and seed storage protein metabolism pathways in rice. Furthermore, expression patterns of *OsMADS1* and other 12 DEGs involved in seed storage protein biosynthesis and regulation in Olr and Nip grains were very similar between qRT-PCR results and mRNA-seq data, which validated the mRNA-seq results (Figure 8; Appendix A).

## 3. Discussion

In this study, we preliminarily reveal that *OsMADS1* regulates grain quality, gene expressions, and regulatory networks of starch and storage protein metabolisms in rice (Figure 9). However, it is very interesting that the functions of *OsMADS1* in regulating grain quality are contradictory and obscure in previous studies. The report of Liu et al. [11] indicates that wild-type *OsMADS1* negatively regulates grain quality in WYJ7, RD23, and LYPJ. By contrast, the report of Yu et al. [13] indicates that *OsMADS1* does not affect grain quality. On the other hand, our results in this study showed that *OsMADS1* positively regulates grain quality (Figure 1 and Figure 3).

However, it is very interesting that the regulatory role of *OsMADS1* in grain quality is seemingly contrary between our results (Figure 1 and Figure 3) and the report of Liu et al. [11]. Firstly, these seemingly contradictory results may be partly derived from the differences in mutation types and resulted in effects on the gene functions between mutated *OsMADS1^lgy3^* and *OsMADS1*^Olr^ alleles. The OsMADS1 protein contains a MADS-box domain located in the N-terminal region of the protein and is responsible for the DNA binding with target genes, I region (Intervening region) and K-box domain (Keratin-like domain) mainly involved in dimerization and protein-protein interactions, and the C-terminal region implicated in transcriptional activation and higher-order complex formation [2]. The mutated *OsMADS1^lgy3^*/*OsLG3b^SLG^* allele encodes an alternatively spliced mutated protein OsMADS1^lgy3^, in which the terminal 37 residues were truncated, and an additional 5 residues were added to its predicted C domain [11,12,13]. This mutation in the C-terminal region of OsMADS1^lgy3^ did not affect the interactions with its interacting proteins, such as DEP1, and the DNA-binding affinity to its target genes, such as *OsMADS55* and *OsPIN1a*, but largely abolished the transactivation activity to its target genes such as *OsMADS55* and *OsPIN1a* [11], which could thereby change and affect the regulatory functions of OsMADS1^lgy3^ to grain development and quality. On the other hand, the I region, K-box domain, and C-terminal region were normal, but the 27th conserved glycine in the MADS-box domain was mutated to aspartic acid in OsMADS1^Olr^ protein (Figure 4D), which could affect its 3D protein structure (Figure 5A–D) and the DNA-binding affinity to its target genes and thereby change and affect the regulatory functions of OsMADS1^Olr^ to spikelet and grain development as well as grain quality. Thus, the differences in mutation types and resulted in effects on the gene functions between mutated *OsMADS1^lgy3^* and *OsMADS1*^Olr^ alleles may partly cause the differences in grain quality between the three NILs of *OsMADS1^lgy3^* (WYJ7*-lgy3-dep1–1*, RD23*-lgy3-gs3* and PA64S/9311*-lgy3-gs3*) and Olr mutant harboring the *OsMADS1*^Olr^.

In the regulation processes of grain quality by *OsMADS1^lgy3^* or *OsMADS1*^Olr^ allele, it needs other cofactors or interacting proteins to form the transcription complex to regulate expressions of a large number of downstream grain quality-related target genes. The *OsMADS1^lgy3^* and *OsMADS1*^Olr^ are individually carried by the three NILs and Olr with different genetic backgrounds. Therefore, it is likely that OsMADS1*^lgy3^* and OsMADS1^Olr^ have some different cofactors, interacting proteins, and downstream target genes, and some genes encoding these cofactors and interacting proteins may also have different spatial and temporal expression patterns in different backgrounds. Therefore, the differences in components and their contents between OsMADS1^lgy3^ and OsMADS1^Olr^ transcription complex in different backgrounds could result in the different effects of gene functions in regulating grain quality between *OsMADS1^lgy3^* and *OsMADS1*^Olr^ alleles.

The three NILs of *OsMADS1^lgy3^* displayed improved grain quality accompanied by normal spikelet fertility, longer grain length, and increased yield. On the other hand, the Olr mutant harboring *OsMADS1*^Olr^ exhibited decreased grain quality and the abnormal grain shape. Furthermore, *OsMADS1^lgy3^* is a useful allele in rice breeding, but *OsMADS1*^Olr^ is not likely to be used in rice breeding directly. These results revealed that compared with the mild *OsMADS1^lgy3^* allele, *OsMADS1*^Olr^ is a more severe allele. In addition, this study and previous results indicate that *OsMADS1* plays critical and dual roles in regulating grain quality and grain shape [10,11,12,13].

Moreover, it also plays balanced regulatory roles in controlling grain shape, which regulates the specification of the lemma and palea and simultaneously acts as an inhibitor of overdevelopment of the lemma, palea, and grain length to maintain the balanced development of rice grains. Overexpression of wild-type *OsMADS1* in Nip resulted in decreased grain length and smaller grain size but suppression of *OsMADS1* expression in Nip caused increased grain length and bigger grain size [11]. The dual and balanced regulatory roles of *OsMADS1* may also exist in the regulation of both grain quality and grain shape in rice and could be partly realized by coordinating the regulation of grain quality and grain shape. Li et al. [19] reported that the appearance quality attribute, grain chalkiness, is positively correlated with the grain shape attribute, grain width. Therefore, the compromise in gene function in the mild mutated allele *OsMADS1^lgy3^* may result in the re-establishment and maintenance of a better balance and homeostasis to regulate grain quality and grain shape, which finally leads to the improvement of grain quality and longer grain length in WYJ7*-lgy3-dep1–1*, RD23*-lgy3-gs3* and PA64S/9311*-lgy3-gs3* NILs. Conversely, a serious impact on gene function in the severely mutated allele *OsMADS1*^Olr^ may damage the normal regulatory roles of *OsMADS1* and break down the balance and homeostasis to regulate grain quality and grain shape, which finally causes the poor grain quality and abnormal grain shape in Olr.

In conclusion, the opposite and seemingly contradictory effects between *OsMADS1^lgy3^* and *OsMADS1*^Olr^ alleles on the grain quality in NIL (SLG), WYJ7*-lgy3-dep1–1*, RD23*-lgy3-gs3*, PA64S/9311*-lgy3-gs3* and Olr can be partly explained and may be caused by the differences of mutation types and resulted in effects on the gene functions of *OsMADS1* alleles, influences of different genetic backgrounds to the gene functions of *OsMADS1* alleles, dual and balanced regulatory roles of *OsMADS1* alleles in regulating grain quality and the grain shape.

Our results showed that *OsMADS1* controls grain quality by regulating starch and seed storage protein metabolism in rice. However, the detailed molecular regulatory mechanism still remains to be explored. Bello et al. [20] reported that two NF-Y type transcription factors, OsNF-YB1 and NF-YC12 (nuclear transcription factor Y subunit C-12), bind to each other and sequentially bind to another Helix-loop-helix transcription factor bHLH144 (basic helix-loop-helix protein 144) to form a heterotrimer complex NF-YB1-YC12-bHLH144, in which NF-YC12 and bHLH144 maintain NF-YB1 stability from the degradation mediated by ubiquitin/26S proteasome, while NF-YB1 directly binds to the G-box (CACGTG) domain of *Wx* (*Waxy, granule-bound starch synthase I gene*) promoter and activates *Wx* transcription, hence to regulate rice grain quality. There is also the possibility that OsMADS1 and other transcription factors can form heterodimerization and transcription complex to jointly regulate transcriptions of some downstream grain quality-related target genes to regulate grain quality. Our results showed that concurrently and overall decreased expressions of *OsMADS1*, *OsNF-YB1*, *OsNF-YC12*, *bHLH144,* and starch biosynthesizing genes in the developing grains of Olr is consistent with the reduced starch content in Olr grains, which partly supports this possibility.

Furthermore, in contrast to the positive regulators of starch biosynthesis, OsNF-YB1, OsNF-YC12, and bHLH144, another transcription factor RSR1 was reported to be a negative regulator of starch biosynthesis, and its expressions were obviously reduced in developing grains of Olr [21] (Figure 6). These results suggest that OsMADS1 may have the dual regulatory roles of activating or/and suppressing expressions of different starch and seed storage protein-related genes in grains by binding or interacting with some of the positive or/and negative regulators such as OsNF-YB1, OsNF-YC12, bHLH144, and RSR1 to accurately regulate starch and seed storage protein biosynthesis, hereafter accurately controlling grain quality. Our results showed that in developing grains of Olr, expressions of starch biosynthesizing and regulating genes were suppressed, but expressions of starch-degrading genes were promoted as a whole, and the seed storage protein biosynthesis regulating genes were suppressed at 1 DAF, 18 DAF, and 24 DAF but promoted at 6 DAF and 12 DAF in general (Figure 6 and Figure 7), which is consistent with this deduction.

On the other hand, RISBZ1 and RPBF encode a rice basic leucine zipper transcription factor and a rice prolamin box binding factor, respectively, which not only interact with each other to regulate expressions of seed storage protein genes through GCN4 Motif and thereby regulate storage protein biosynthesis but also regulate expressions of some of the starch biosynthesizing genes such as *Wx* [22,23]. Thus, similar to RISBZ1 and RPBF, there is also the possibility that OsMADS1 can regulate expressions of both starch, seed storage protein metabolizing, and related genes to collaboratively regulate starch and seed storage protein metabolism, thereby controlling grain quality.

## 4. Materials and Methods

### 4.1. Plant Materials and Growth Conditions

The ‘Olr (Oat-like rice)’ is a stably hereditable and recessive mutant caused by a spontaneous mutation, which was originally discovered in the paddy field in 2001 [10]. The *japonica* cv. Nip (Nipponbare) was used as a control of Olr in the comparative phenotypic and transcriptomic analyses during grain development because its original wild type is unknown. Nip was also used as the transgenic recipient and wild-type control in the *OsMADS1* seed-specific RNAi (p*OsTip3*::*OsMADS1*-RNAi) transgenic experiment. The detailed method for constructing the p*OsTip3*::*OsMADS1*-RNAi transgenic lines was described in the study of Li et al. [10]. Olr, Nip, the F_2_ and F_3_ progeny plants derived from a cross combination of Olr/Nip were used for the determination of starch and protein contents in grains. Field-grown rice plants were grown under normal paddy conditions during natural growing seasons in an isolated experimental plot in Chengdu city, Sichuan province of China.

### 4.2. Measurement of Starch, Amylose, Amylopectin, and Protein Content

Brown rice from Nip, Olr, F_2,_ and F_3_ progeny plants of Olr and Nip were milled into fine flour by using a flour mill. The fine flour from each of the samples was subsequently used for the determination of the starch, amylose, amylopectin, and protein contents. Starch content, amylose content, and amylopectin contents were determined by using the starch assay kits Megazyme K-TSTA and K-AMYL (Megazyme, Wicklow, Ireland) as described by Wei et al. [24]. The protein content of the samples was determined following the method described by Kang et al. [25].

### 4.3. Vector Construction and Rice Transformation

For the construction of the p*Ubi*::*OsMADS1* vector used to express the wild-type *OsMADS1* gene in Olr to rescue its poor grain quality, the wild-type *OsMADS1* cDNA was cloned from Nip and was inserted into the binary vector pCUbi1390 [26] at the BamH I site and the Spe I site. Then, the p*Ubi*::*OsMADS1* vector was transformed into the Olr calli to obtain T_0_ transgenic plants by *Agrobacterium tumefaciens* (EHA105)-mediated transformation [27]. The corresponding primers developed for constructing the p*Ubi*::*OsMADS1* vector are listed in Appendix A.

### 4.4. Transcriptome Samples Collection and Treatment

Nip and Olr grains at representative time points during grain development consisting of 1 DAF, 6 DAF, 12 DAF, 18 DAF, and 24 DAF were collected from field-grown rice plants. For all grain samples at each time point, three biological replicates were harvested from Nip and Olr plants. Harvested samples were immediately frozen in liquid nitrogen and stored at −80 °C. Then all samples frozen by drikold were shipped to Basebio Biotechnology Co., Ltd. (Chengdu 610041, China) for sample quality testing, cDNA library construction, transcriptome sequencing, and initial data analysis.

### 4.5. Transcriptome Data Analysis

The sequencing quality of obtained raw data with the fastq format of each sample was assessed using the FastQC tool (https://www.bioinformatics.babraham.ac.uk/projects/fastqc/, accessed on 27 April 2023). Subsequently, the clean reads of each sample with high quality were aligned to the rice reference genome sequence (Nipponbare rice genome IRGSP-1.0) using the Tophat2 software (v2.1.1, Center for Bioinformatics and Computational Biology, University of Maryland, College Park, MD, USA) [28]. For transcript quantification, HTSeq-count script (v2.0.2, https://pypi.org/project/HTSeq/, accessed on 27 April 2023) was then used to obtain the calculated reads count and FPKM (fragments per kilobase of transcript per million mapped reads) value for every single transcript of every gene in each sample. To test the correlation of different samples and the similarity of biological replicates, the PCA (Principal Component Analysis) and sample-to-sample correlation analysis were used. DEGs testing was analyzed using DESeq R packages according to the manual. Raw count data was prepared by custom perl script based on the results of eXpress software and was imported into the DESeq framework. The FDRs (False Discovery Rates) were controlled using the Benjamini–Hochberg method at an FDR of 5% [29]. For analyzing the overlapping DEGs, Venn diagram analysis was employed [30]. Functional enrichment analysis of DEGs is based on the hypergeometric test. The obtained DEGs between Olr and Nip grain samples were annotated by the databases of GO and KEGG and subsequently subjected to GO and KEGG significant enrichment analysis to identify starch and storage protein-related GO functional categories and KEGG pathways [31,32]. SOMs were constructed to identify and compare the expression patterns of DEGs between Olr and Nip grains by using SOM R packages. Subsequently, temporal expression profiles of the obtained robust clusters were constructed based on the extracted characteristic values of the gene expression levels in the robust clustering maps by the SOMs analysis [33].

### 4.6. Grain Appearance Quality Analysis and Microscope Observation

Mature grains were harvested and naturally dried, then stored at room temperature. Grains of Nip and Olr were dehulled using a rice huller or manually. Then brown rice of Nip and Olr was ground into white rice (endosperms) using a rice polisher. The appearance quality of these white rice, including the percentage of endosperms with chalkiness and degree of endosperm chalkiness, was analyzed using a rice quality analyzer (TPMZ-A, Zhejiang Top Cloud-agri Technology Co., Ltd., Hangzhou, China).

Brown rice grains of Nip and Olr were transversely cut in the middle using a dissecting blade. Then the corresponding half-brown rice grains were mounted on stubs, and the transverse sections of these samples were first observed under a dissecting microscope (SMZ745T, Nikon, Tokyo, Japan). Then each of the same observed samples was mounted on a stub, and the transverse section was coated with gold and subsequently observed under an SEM (Inspect S50, FEI Company, Hillsboro, OR, USA) with an accelerating voltage of 20.00 kV. The transverse sections of the brown rice grains with or without chalkiness and the corresponding starch granules were observed and photographed under the SEM.

### 4.7. Mapping-by-Sequencing the Causal Gene of Grain Quality Phenotype in Olr

For mapping and cloning the causal gene through the mapping-by-sequencing method, we initially constructed an F_2_ segregating population by a cross between Olr (maternal plant) and Nip (paternal plant) [10]. Then, the maternal bulk DNA (OM-bulk) and paternal bulk DNA (NP-bulk) of the parental plants were prepared by extracting and mixing equal amounts (500 ng total DNA) of DNA samples from nine individual plants of Olr and Nip. In addition, DNA samples were extracted from 152 individual plants showing Olr grain shape (O-bulk), and another 152 individual plants showing normal grain shape (N-bulk) in the F_2_ segregating population; and the two groups of DNA samples were mixed with the equal amount (500 ng total DNA) of the 152 DNA samples in each group, and finally to be served as the O-bulk and N-bulk, respectively. Then, the DNA samples frozen by drikold were shipped to Shanghai OE Biotech Inc. (Shanghai 201100, China) for DNA library construction, high throughput sequencing, and initial data analysis.

Pooled libraries with insert sizes of around 350–500 bp were prepared and sequenced on a HiSeq X™ Ten system (Illumina, San Diego, CA, USA) by using 150-base paired-end sequencing. In addition, the short reads obtained from the four samples were aligned to the reference genome of the *japonica* cv. Nip (IRGSP-1.0) using BWA software (version 0.7.17, Wellcome Trust Sanger Institute, Wellcome Trust Genome Campus, Cambridge, UK) [34]. Alignment files were converted to SAM/BAM files using SAMtools and applied to the SNP-calling.

Unique SNP sites were filtered (ratio ≥ 0.8) in both maternal bulk (OM-bulk) and paternal bulk (NP-bulk) by comparing the two bulks. The intersections among unique SNP sites from the two maternal & paternal bulks and SNP sites of the progeny bulks (O-bulk and N-bulk) were taken. Then the SNP index of the screened SNP sites was calculated. SNP-index = Number of the reads with the SNP derived from the maternal plant, Olr/(Number of the reads with the SNP derived from the maternal plant, Olr + Number of the reads with the SNP derived from the paternal plant, Nip). If SNP-index = 0, it indicates that all the reads of the SNP were derived from the paternal plant, Nip; but if SNP-index = 1, it indicates that all the reads of the SNP were derived from the maternal plant, Olr.

Sliding window analysis was subsequently applied to calculate the SNP-index of O-bulk and N-bulk and to calculate the ∆(SNP-index) between O-bulk and N-bulk (O-bulk—N-bulk) based on a 1 Mb interval with a 10 kb sliding window. The SNP-index plots (O-bulk and N-bulk), ∆(SNP-index) plot (O-bulk—N-bulk), as well as the corresponding linkage maps, were simultaneously drawn and constructed. For the ∆(SNP-index) plot, the 99% (*p* < 0.01) and 95% (*p* < 0.05) statistical confidence intervals were set under the null hypothesis of no causal gene, or QTL was identified within the confidence intervals.

Location and further screening of the candidate causal SNPs and corresponding candidate genes were based on the criteria that SNP-index in paternal bulk (PN-bulk) is 0.00, SNP-index in maternal bulk (MO-bulk) is no less than 0.85, SNP-index in O-bulk is 1.00, SNP-index in N-bulk ranges from 0.30 to 0.40, the corresponding Δ(SNP-index) between O-bulk and N-bulk ranges from 0.60 to 0.70, the SNP is located within the gene which either caused missense mutation or stop codon of the encoding protein. The screened candidate causal SNPs, positions, corresponding candidate genes, location of mutations within genes, effects on amino acid changes, and predicted gene functions were annotated using SnpEff software (version 5.1, https://pcingola.github.io/SnpEff/, accessed on 27 April 2023).

### 4.8. Immunoblot Analysis

The proteins were extracted from the flag leaves, florets, and seeds of rice using the Lysis Buffer [50 mM Tris-HCl, pH 7.5; 150 mM NaCl; 1 mM EDTA; 10% glycerol; 2 mM Na_3_VO_4_; 25 mM glycerophosphate; 10 mM NaF; 0.05–0.1% Tween20; 1 × Protease Inhibitor Cocktail (MedChemExpress, Shanghai, China); 1 mM PMSF]. The obtained protein lysates were immunoblotted with the OsMADS1 antibody (Catalog No.: A20328; ABclonal Technology Co., Ltd., Wuhan, China), and the abundance of HSP82 protein detected by the HSP82 antibody (Beijing Protein Innovation Co., Ltd., Beijing, China) was used as an internal control.

### 4.9. Three-Dimensional Protein Structure Prediction

The 3D protein structure prediction of wild-type OsMADS1 and mutated OsMADS1^Olr^ proteins was performed using the PyMOL molecular visualization program (https://pymol.org/2/, accessed on 27 April 2023).

### 4.10. Protein Subcellular Localization

The full-length CDS (coding sequence) of wild-type *OsMADS1* and mutated *OsMADS1*^Olr^ was amplified from the Nipponbare and Oat-like rice cDNA using the PCR primers listed in Appendix A. The verified CDS fragments of *OsMADS1* and *OsMADS1*^Olr^ were cloned in the frame in front of the eGFP CDS in the pJIT163-P*35S*::*eGFP* vector to construct the pJIT163-P*35S*::*OsMADS1-eGFP* and pJIT163-P*35S*::*OsMADS1*^Olr^*-eGFP* transient overexpression vector driven by the 35S promoter. P*35S*::*OsMADS1-eGFP* or pJIT163-P*35S*::*OsMADS1*^Olr^*-eGFP* vector was transferred into and transiently expressed in rice protoplasts using the methods as described previously [35,36]. The eGFP fluorescent signals were detected using a confocal laser-scanning microscope (Leica TCS SP8, Wetzlar, Germany).

### 4.11. qRT-PCR Analysis

To verify the Illumina comprehensive transcriptomic data, the same batch of samples for preparation of the Illumina sequencing libraries was used to analyze the expression profiles of storage protein-related genes in Nip and Olr by qRT-PCR analysis. The qRT-PCR analysis, including cDNA synthesis, was performed following the method described by Li et al. (2020) [10]. The *OsActin* gene in rice was used as an endogenous control. The relative gene expression levels were calculated by using the 2^−ΔΔCT^ algorithm. The primers used in the qRT-PCR analysis are listed in Appendix A.

## Figures and Tables

**Figure 1 ijms-24-08017-f001:**
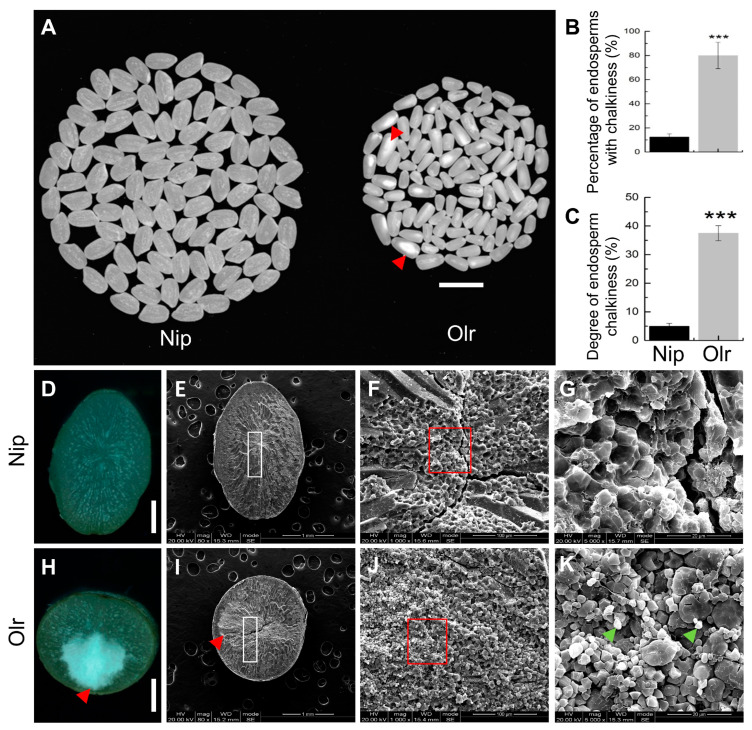
Comparative analysis of appearance quality in endosperms and starch granules in brown rice between Nip and Olr. (**A**–**C**) Appearance quality of Nip and Olr endosperms; (**A**) appearance of endosperms, (**B**) percentage of endosperms with chalkiness, (**C**) degree of endosperm chalkiness. (**D**–**K**) Comparative observation and analysis of chalkiness (**D**,**E**,**H**,**I**) and starch granules (**F**,**G**,**J**,**K**) in the transverse sections of Nip (**D**–**G**) and Olr (**H**,**K**) brown rice by dissecting microscope (**D**,**H**) and SEM (**E**–**G**,**I**–**K**). (**F**,**G**,**J**,**K**) Comparison of starch granules between the chalkiness area in the Olr brown rice (**J**,**K**) and the corresponding area without chalkiness of Nip (**F**,**G**). (**F**,**G**) Enlarged view of the white rectangle in (**E**) (Nip) and (**I**) (Olr). (**G**,**K**) Further enlarged view of the red rectangle in (**F**) (Nip) and (**J**) (Olr), which shows enlarged starch granules; Small, irregular, and loosely packed starch granules are indicated by green arrowheads in (**K**). Red arrowheads indicate chalkiness (**A**,**H**,**I**). SEM: Scanning electron microscope. Bars: 1 cm (**A**), 1 mm (**D**,**E**,**H**,**I**), 100 μm (**F**,**J**), 20 μm (**G**,**K**). Data are given as mean ± SD [*n* = 3 replicates (each replicate consists of about 200 endosperms in (**B**,**C**)]. Student’s *t*-test: *** *p* < 0.001.

**Figure 2 ijms-24-08017-f002:**
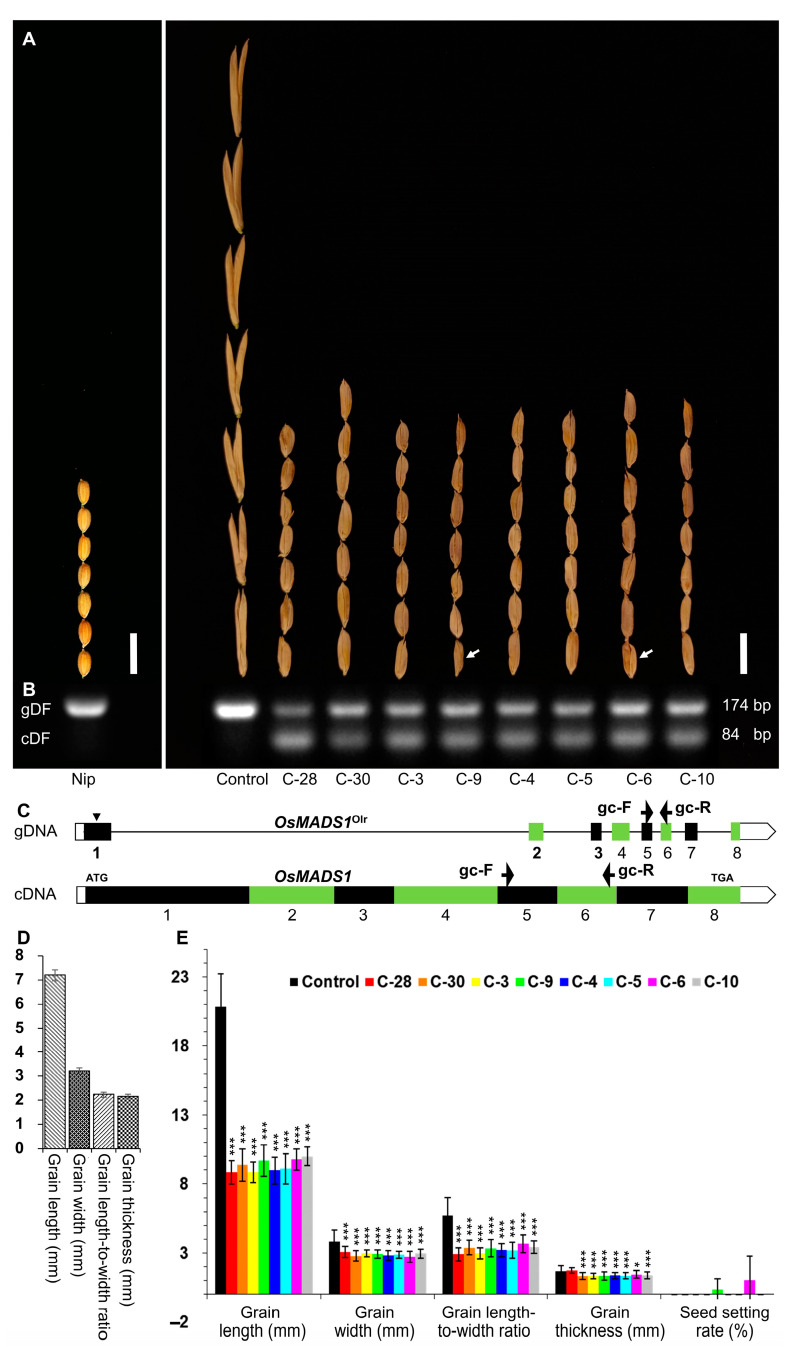
Rescue of the Olr grain shape by complementation of the *OsMADS1* gene. (**A**) Comparison of grain shape among Nip, T_0_ negative control plant (Control, Olr genetic background) and positive plants (C-28, C-30, C-3, C-9, C-4, C-5, C-6, and C-10). The T_0_ negative control plant doesn’t harbor the exogenous wild-type *OsMADS1* gene (cDNA) because the exogenous p*Ubi*::*OsMADS1* vector failed to be transformed into the calli of Olr by *Agrobacterium tumefaciens* (EHA105)-mediated transformation. White arrows indicate fertile seeds. Bars: 10 mm. (**B**) PCR confirmation of T_0_-positive transgenic plants. The gDF and cDF indicate the DNA fragments amplified from mutated *OsMADS1*^Olr^ genomic DNA (gDNA) of Olr and cDNA fragments amplified from wild-type *OsMADS1* cDNA derived from Nip (cDNA), with the gc-F/gc-R primer pair as shown in (**C**). (**C**) PCR primers (Appendix A) used to identify T_0_-positive transgenic plants in (**B**). Black arrowhead indicates the mutation site in *OsMADS1*^Olr^ gene. Black arrows indicate gc-F and gc-r primers. ATG and TGA represent the start codon and stop codon, respectively. Exons are shown as black or green closed boxes; introns are shown as black straight lines; and UTRs (untranslated regions) are marked with open boxes. (**D**,**E**) Comparison of grain length, grain width, grain length-to-width ratio, grain thickness, and seed setting rate among Nip (**D**), T_0_-negative control plant (Control), and positive plants (C-28, C-30, C-3, C-9, C-4, C-5, C-6, and C-10) (**E**). Data in (**D**,**E**) is given as mean ± SD [*n* = 30 for grain length, grain width, grain length-to-width ratio, and grain thickness; *n* = 10 for seed setting rate]. Student’s *t*-test: * *p* < 0.05, *** *p* < 0.001.

**Figure 3 ijms-24-08017-f003:**
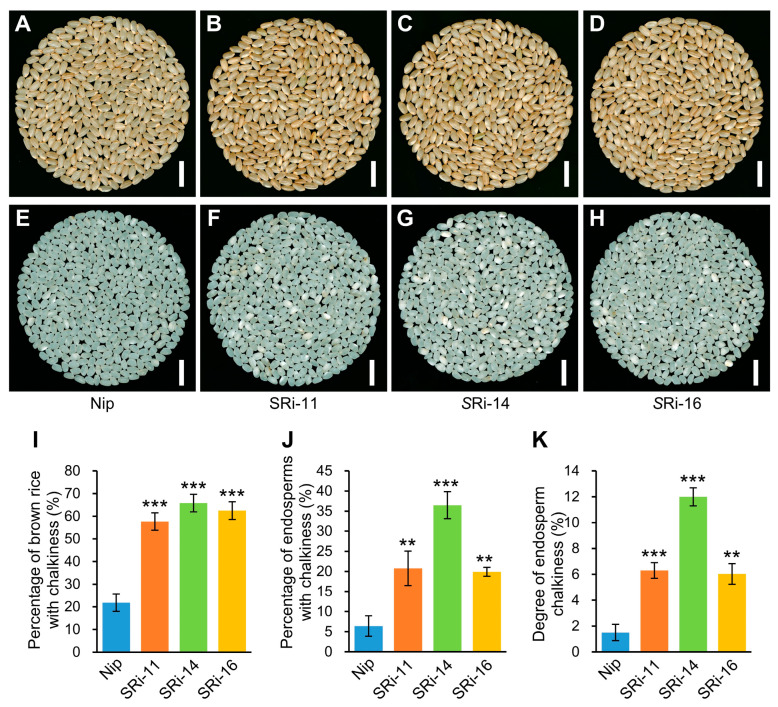
Decreased grain appearance quality of *OsMADS1* seed-specific RNAi lines. (**A**–**H**) Comparisons of grain chalkiness of brown rice (**A**–**D**) and endosperms (**E**–**H**) between homozygous T_3_ *OsMADS1* seed-specific RNAi (p*OsTip3*::*OsMADS1*-RNAi) lines (SRi11, SRi14, and SRi16) and wild-type plants (Nip). T_4_ seeds generated from the corresponding T_3_ plants were analyzed. Bars: 10 mm (**A**–**H**). (**I**–**K**) Comparisons of percentage of brown rice with chalkiness (**I**), percentage of endosperms with chalkiness (**J**), and degree of endosperm chalkiness (**K**) between *OsMADS1* seed-specific RNAi lines and wild-type plants. Data are given as mean ± SD [*n* = 6 × 100 brown rice grains for each line in (**I**); *n* = 3 × about 200 endosperms for each line in (**J**,**K**)]. Student’s *t*-test: ** *p* < 0.01, *** *p* < 0.001.

**Figure 4 ijms-24-08017-f004:**
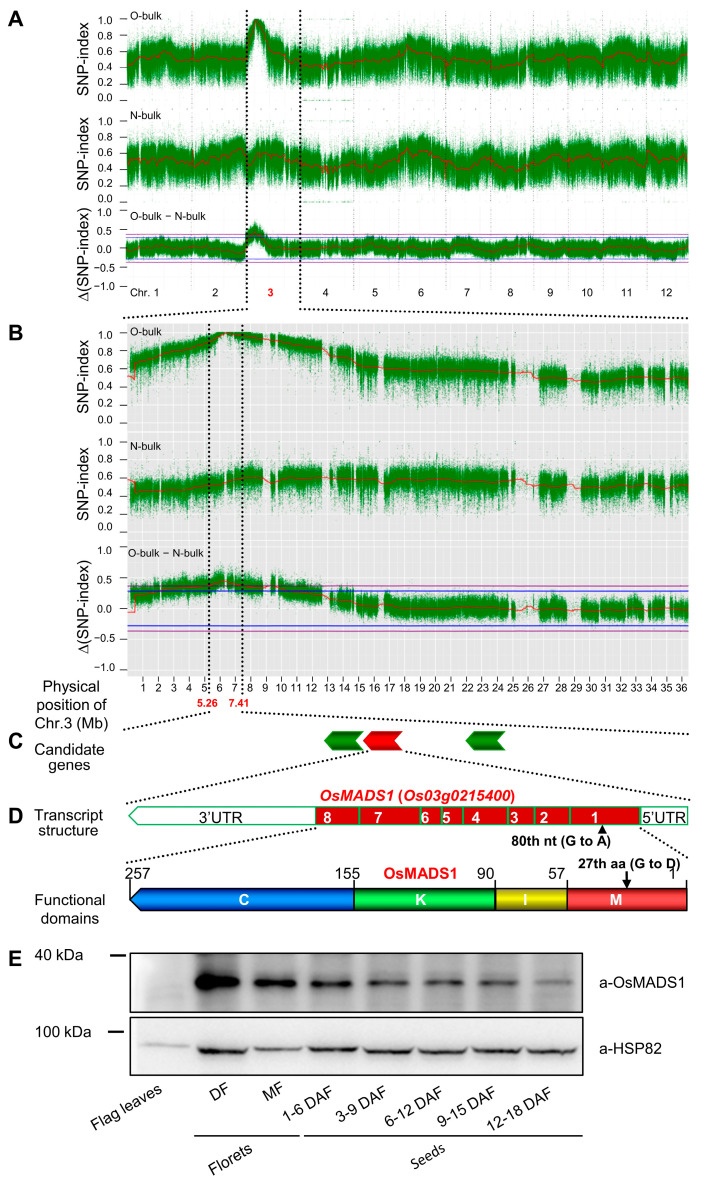
Mapping-by-sequencing the causal gene of grain quality phenotype in Olr. (**A**,**B**) Mapping of the causal gene in the 5.26~7.41 Mb interval of chromosome 3. Dense and green spots in the upper and middle plots of (**A**,**B**) represent the single nucleotide polymorphism (SNP)-index of O-bulk (bulk DNA sample extracted from 152 F_2_ individual plants showing Olr phenotypes) and N-bulk (bulk DNA sample extracted from 152 F_2_ individual plants showing normal phenotypes), respectively; while dense and green spots in the lower plots of (**A**,**B**) represent the ∆(SNP-index) between O-bulk and N-bulk (O-bulk—N-bulk). Red curved lines mean the average SNP-index (O-bulk and N-bulk) or ∆(SNP-index) calculated based on a 1 Mb interval with a 10 kb sliding window. Purple and blue straight lines in the ∆(SNP-index) plots (**A**,**B**) mark the 99% (*p* < 0.01), and 95% (*p* < 0.05) statistical confidence intervals under the null hypothesis of no gene or QTL (quantitative trait locus) was identified within the confidence intervals. Chr. indicates chromosome. (**C**,**D**) Schematic representation of transcript structure of *OsMADS1* gene (**C**) and the encoding protein (**D**). Closed red boxes and open boxes indicate exons and UTRs (untranslated regions), respectively; black arrowhead represents the 80th nucleotide substitution site in the mutated *OsMADS1*^Olr^ allele; nt means nucleotide (**C**). M, I, K, and C indicate the MADS-box domain, I region, K-box domain, and C-terminal region of OsMADS1 protein, respectively; black arrow indicates the 27th amino acid substitution site in the MADS-box of the mutated OsMADS1^Olr^ protein; aa represents amino acid (**D**). (**E**) Protein expression pattern of OsMADS1 by immunoblot analysis. Protein lysates from flag leaves, florets, and seeds of Nipponbare were immunoblotted with the OsMADS1 antibody (a-OsMADS1). An abundance of HSP82 (heat shock protein 82) detected by the HSP82 antibody (a-HSP82) was used as an internal control. Protein molecular weight markers are denoted by kDa. DF, developing florets; MF, mature florets; DAF, days after fertilization.

**Figure 5 ijms-24-08017-f005:**
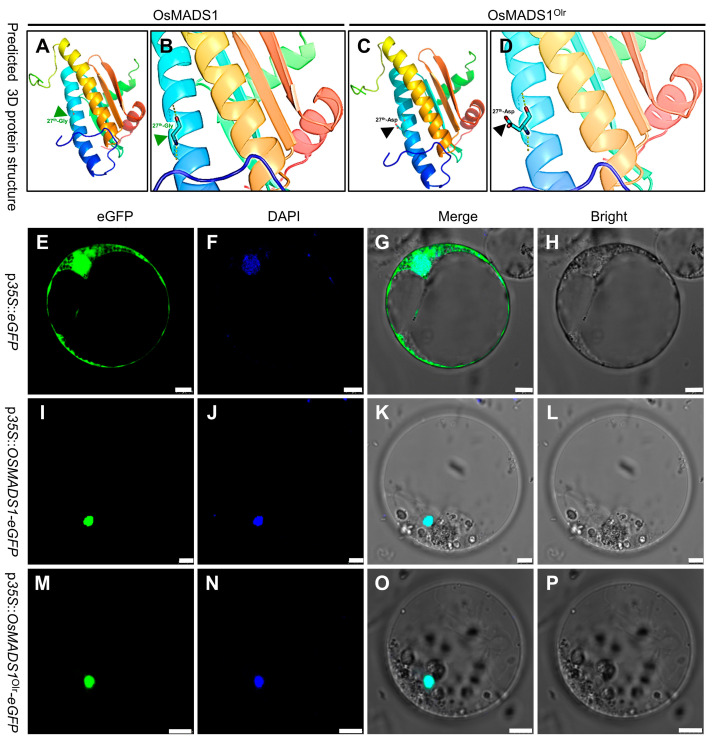
Three-dimensional (3D) structure prediction and subcellular localization of OsMADS1 and OsMADS1^Olr^ proteins. (**A**,**C**) 3D protein structure prediction of wild-type OsMADS1 (**A**) and mutated OsMADS1^Olr^ (**C**). (**B**,**D**) Enlarged view of the 27th amino acid site in the MADS-box domain for target DNA binding of OsMADS1 (**B**) and OsMADS1^Olr^ (**D**). Blue arrowhead (**A**,**B**) and black arrowhead (**C**,**D**) indicate the wild-type glycine (27th-Gly) and mutated aspartic acid (27th-Asp) at the 27th amino acid site in the MADS-box domain of OsMADS1 and OsMADS1^Olr^, respectively. (**E**–**P**) Subcellular localization of OsMADS1-eGFP and OsMADS1^Olr^-eGFP fusion proteins in rice protoplasts. The nuclear affinity dye DAPI (4′,6-diamidino-2-phenyl-indole) was used as a nuclear marker. Bars: 5 μm (**E**–**P**).

**Figure 6 ijms-24-08017-f006:**
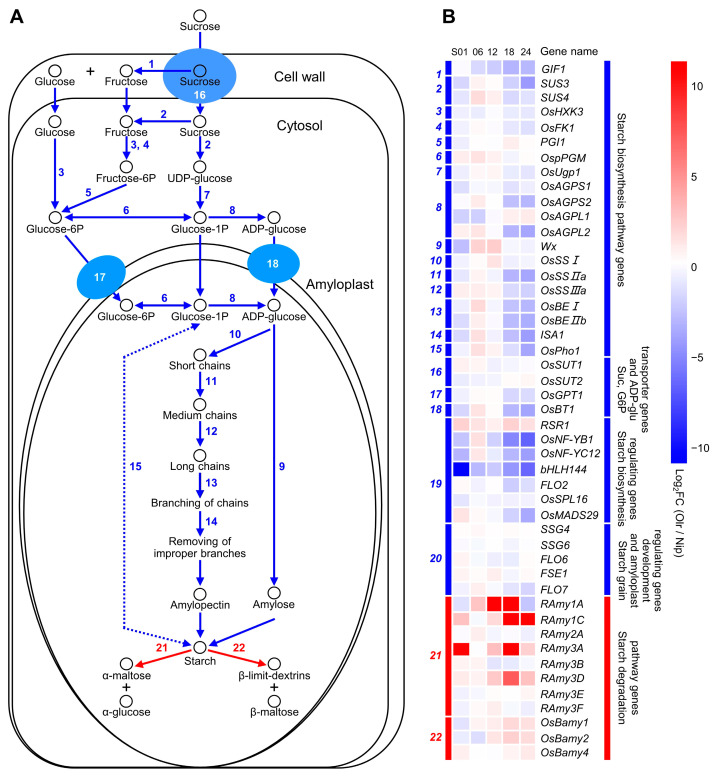
Starch metabolism pathways and related genes expression analysis between Olr and Nip grains. (**A**) Starch biosynthesis and degradation pathways in rice grains. Schematic diagram shows starch biosynthesis and degradation pathways in endosperm cells. Solid and dotted arrows indicate the steps of verified and predicted enzyme-catalyzed reactions, respectively. Single-headed and double-headed arrows indicate irreversible and reversible reactions, respectively. Numbers in Roman type beside the arrows represent the enzyme (enzymes) encoding by corresponding gene (genes) catalyzes (catalyze) this step of reaction. Steps of enzyme-catalyzed reactions and corresponding enzymes in the starch biosynthesis and degradation pathways are marked in blue and red, respectively. Blue ellipses and attached arrows show sucrose, glucose-6P, and ADP-glucose (ADP-glu) transporters encoded by corresponding genes in the starch biosynthesis pathways, and corresponding transport steps, respectively. (**B**) Expression analysis of starch biosynthesis, degradation, and related regulating genes between Olr and Nip during grain development. Blue and red bars encoded by italic numbers marked by the same colors in the left panel of the heatmap indicate the encoding gene (genes) of the enzyme (enzymes) which catalyzes (catalyze) the corresponding step (steps) of enzyme-catalyzed reactions in the starch biosynthesis and degradation pathways in (**A**), as well as the regulating genes of starch biosynthesis, starch granules, and amyloplast development. Names of these genes are listed on the right side of the heatmap. Blue and red bars in the right panel of the heatmap indicate these genes are divided into the starch biosynthesis pathway genes, sucrose, glucose-6P and ADP-glucose transporter genes, starch biosynthesis, starch granules and amyloplast development regulating genes, and starch degradation pathway genes, respectively. In addition, detailed information on these genes, including accession numbers, encoding enzymes (or proteins), and catalytic reactions (or regulatory functions), are summarized in Appendix A. Color grids of genes in the heatmap and color scale at the upper right indicate the value of log2 fold change (FC) of genes between Olr and Nip [Log2FC (Olr/Nip)], thereby red and blue color representing upregulated and downregulated expression of genes in Olr compared with that of Nip. S01, S06, S12, S18, and S24 indicate 1 DAF, 6 DAF, 12 DAF, 18 DAF, and 24 DAF grains, respectively. DAF: days after fertilization.

**Figure 7 ijms-24-08017-f007:**
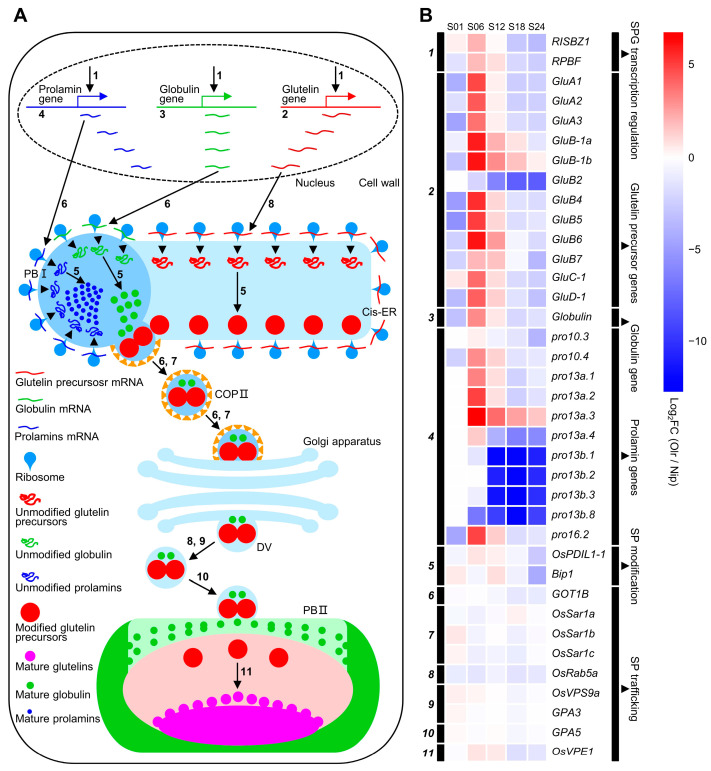
Seed storage protein biosynthesis and accumulation pathways, and related genes expression analysis between Olr and Nip grains. (**A**) Seed storage protein biosynthesis, processing, trafficking, and regulating pathways in rice grains. Arrows indicate the seed storage protein biosynthesis, processing, trafficking, accumulating, and regulating steps. Numbers in Roman type beside the arrows represent the protein (proteins) encoding by corresponding gene (genes) in (**B**), which regulates (regulate) this step. Cis-ER: cisternal endoplasmic reticulum; COP II: coat protein complex II; DV: dense vesicle; PB I: protein body I; PB II: protein body II. (**B**) Expression analysis of seed storage protein biosynthesis, trafficking, and regulating genes between Olr and Nip during grain development. Black bars encoded by italic numbers in the left panel of the heatmap indicate the encoding gene (genes) of the protein (proteins) which regulates (regulate) transcriptions of seed storage protein genes, controls (control) biosynthesis, processing, trafficking, or accumulation of seed storage protein in (**A**). Names of these genes are listed on the right side of the heatmap. Black bars in the right panel of the heatmap indicate these genes are divided into transcription factor genes which regulate seed storage protein genes transcription, glutelin precursor, globulin, and prolamin encoding genes, as well as seed storage protein modifying and trafficking genes. Detailed information on these genes, including accession numbers, encoding proteins, and regulatory functions, is summarized in Appendix A. Color grids of genes in the heatmap and color scale at the upper right indicate the value of log2 FC of genes between Olr and Nip [Log2 FC (Olr/Nip)], thereby red and blue color representing upregulated and downregulated expression of genes in Olr compared with that of Nip. S01, S06, S12, S18, and S24 indicate 1 DAF, 6 DAF, 12 DAF, 18 DAF, and 24 DAF grains, respectively. DAF: days after fertilization; SP: seed storage protein; SPG: seed storage protein gene.

**Figure 8 ijms-24-08017-f008:**
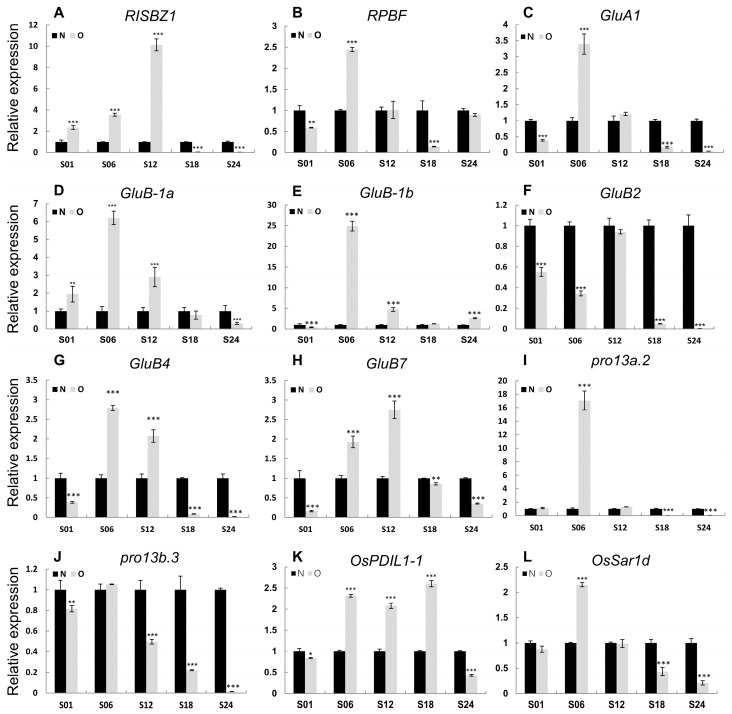
Validation of the comprehensive transcriptomic data by qRT-PCR analysis. (**A**,**I**) Expression patterns of representative transcriptional regulatory genes of storage protein (**A**,**B**), glutelin encoding genes (**C**–**H**), prolamin encoding genes (**I**,**J**), storage protein modification (**K**), and trafficking (**L**) genes in Olr and Nip grains. The value of *OsActin* mRNA was used as an internal control for data normalization, and the expression levels of genes in Nip at S01 time point were set as 1. N: Nip; O: Olr. S01, S06, S12, S18, and S24 indicate 1 DAF, 6 DAF, 12 DAF, 18 DAF, and 24 DAF grains, respectively. DAF: days after fertilization. Data are given as mean ± SD of three replicates. Student’s *t*-test: * *p* < 0.05, ** *p*< 0.01, *** *p* < 0.001.

**Figure 9 ijms-24-08017-f009:**
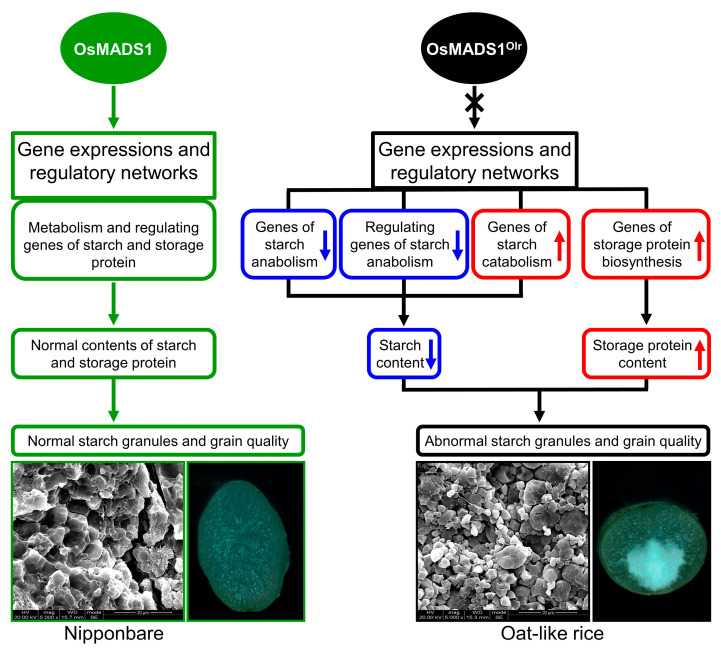
A proposed model depicting how *OsMADS1* regulates grain quality, gene expressions, and regulatory networks of starch and storage protein metabolisms in rice.

**Table 1 ijms-24-08017-t001:** Starch and protein contents of brown rice from Nip, Olr, and their progeny plants.

Sample Name	Starch Content (%)	Amylose Content (%)	Amylopectin Content (%)	Protein Content (%)
Nip	86.66 ± 0.74	20.43 ± 0.15	66.23 ± 0.81	9.31 ± 0.05
Olr	77.55 ± 1.21 ***	19.50 ± 0.38 *	58.05 ± 0.83 ***	14.13 ± 0.07 ***
F_2_ plants showing normal phenotypes	88.10 ± 2.65	17.23 ± 2.89	70.87 ± 0.30	8.79 ± 1.78
F_2_ plants showing Olr phenotypes	82.03 ± 0.15 ***	15.55 ± 0.08	66.47 ± 1.50 ***	13.11 ± 0.63 ***
F_3_ plants showing normal phenotypes	88.32 ± 0.13	17.96 ± 0.09	70.36 ± 0.20	7.56 ± 0.05
F_3_ plants showing Olr phenotypes	75.71 ± 1.53 ***	15.95 ± 0.04 ***	59.76 ± 1.49 ***	14.90 ± 0.03 ***

F_2_ and F_3_ plants were derived from a cross between Olr and Nip. Olr phenotypes indicate poor grain quality with chalky endosperms and abnormal grain shape with lower 1000-brown rice weight and slower grain-filling rate exhibited in Olr grains; normal phenotypes indicate normal grain quality and grain shape in Nip grains. Data are given as mean ± SD of the percentage of starch content, amylose content, amylopectin content, and protein content in the dry basis of each sample. Student’s *t*-test: * *p* < 0.05, *** *p* < 0.001.

## Data Availability

The raw reads of RNA-seq in this study are available under the accession number PRJNA798442 at the National Center for Biotechnology Information (NCBI).

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
