# Peer review of "OsMADS1 Regulates Grain Quality, Gene Expressions, and Regulatory Networks of Starch and Storage Protein Metabolisms in Rice"

_ijms, 2023, doi:10.3390/ijms24098017_

Round 1
Reviewer 1 Report
The manuscript "OsMADS1 regulates grain quality, gene expressions and regulatory networks of starch and storage protein metabolisms in rice" provides compelling evidence for the role of MADS1 in grain development and maturation. This paper follows previous evidence suggesting a role for MADS-box family members in shaping grain development content besides inflorescence and spikelets, reinforcing this notion showing that one more gene belonging to this family is involved. The relevant literature is well reported in the background, and the methods employed in the work are sound, well described, and appropriate for the goal of the study. The gene expression analyses well characterize the impact of MADS1 on the grain and complement well the mapping by sequencing approach used to pinpoint the role of this gene in Olr phenotypes. Concluding, I think the paper should be accepted. I also commend the writing of the paper, which is very good and left me unable to find any significant correction nor point that can improved substantially.
A minor comment is only that some of the terminology is well described in the method section, but referred to earlier in the text without being properly introduced. This is the case for SNPindex, for example. While readers familiar with mapping by sequencing can guess easily what this is, for others it might be harde to follow the paper. This is a minor aspect however, which could be improved easily in a final version of the manuscript.
Author Response
Responses to the comments of Reviewer 1
We highly appreciate your generous encouragement! Your comments are highly insightful and illuminating, and have enabled us to greatly improve the quality of our manuscript. We highly appreciate your kind help and valuable contributions to our manuscript! In the following pages are our point-by-point responses to your comments.
Reviewer 1
Comments and Suggestions for Authors
The manuscript "OsMADS1 regulates grain quality, gene expressions and regulatory networks of starch and storage protein metabolisms in rice" provides compelling evidence for the role of MADS1 in grain development and maturation. This paper follows previous evidence suggesting a role for MADS-box family members in shaping grain development content besides inflorescence and spikelets, reinforcing this notion showing that one more gene belonging to this family is involved. The relevant literature is well reported in the background, and the methods employed in the work are sound, well described, and appropriate for the goal of the study. The gene expression analyses well characterize the impact of MADS1 on the grain and complement well the mapping by sequencing approach used to pinpoint the role of this gene in Olr phenotypes. Concluding, I think the paper should be accepted. I also commend the writing of the paper, which is very good and left me unable to find any significant correction nor point that can improved substantially.
Point 1: A minor comment is only that some of the terminology is well described in the method section, but referred to earlier in the text without being properly introduced. This is the case for SNPindex, for example. While readers familiar with mapping by sequencing can guess easily what this is, for others it might be harde to follow the paper. This is a minor aspect however, which could be improved easily in a final version of the manuscript.
Response: We highly appreciate these very important and valuable comments and suggestions given by the reviewer. We have rechecked the whole manuscript and made the following revisions according to the suggestions from the reviewer.
(1) To make the SNP-index easier for the reader to understand, we have added calculation formula of the SNP-index in the Materials and Methods 4.7. as “SNP-index = Number of the reads with the SNP derived from the maternal plant, Olr / (Number of the reads with the SNP derived from the maternal plant, Olr + Number of the reads with the SNP derived from the paternal plant, Nip).” (lines 880-882).
Furthermore, we have cited the Materials and Methods 4.7. in the text of Results 2.4. as “Screened SNPs (single nucleotide polymorphisms) between Olr and Nip were used to calculate the SNP-index of O-bulk and N-bulk, and the corresponding ∆(SNP-index) between the two bulks. Then, linkage maps between SNPs and the candidate causal gene in the 12 chromosomes of rice were constructed by the SNP-index and ∆(SNP-index), and were subsequently used to identify the location of the candidate causal gene (Materials and Methods 4.7.).” (lines 274-279)
(2) we have added the corresponding full names or annotations of the abbreviations of ADP (adenosine-5′-diphosphate) (lines 57-58), UDP (uridine 5′-diphosphate) (line 62), glucose-6P (glucose-6-phosphate) (line 63), glucose-1P (glucose-1-phosphate) (line 66), RISBZ1 (rice basic leucine zipper factor) (lines 85-86), RPBF (rice prolamin box binding factor) (line 86), GOT1B (Golgi transport 1) (line 88), OsRab5a (a small GTP-binding protein) (line 90), OsPDIL1-1 (a protein disulphide isomerase-like enzyme) (line 93), Bip1 (binding protein 1) (line 93), OsSar1a, OsSar1b and OsSar1c (secretion-associated, Ras-related protein 1a, 1b and 1c) (lines 97-98), OsVPS9a (vacuolar protein sorting-associated protein 9a) (line 100), GPA3 (glutelin precursor accumulation 3) (line 100), OsMADS29-RNAi (OsMADS29-RNA interference) (line 116), G-protein (guanine nucleotide-binding protein) (line 131), WYJ7 (Wuyungeng No. 7) (line 139), PA64S (Peiai 64S) (line 140), LYPJ (Liangyoupeijiu, PA64S / 9311) (line 155), SNPs (single nucleotide polymorphisms) (lines 274-275), QTL (quantitative trait locus) (line 300), HSP82 (heat shock protein 82) (line 309), OsMADS1-eGFP (OsMADS1-enhanced green fluorescent protein) (lines 329-330), OsTip3 (tonoplast intrinsic protein 3) (line 393), OsAGPS1, OsAGPS2 (ADP-glucose pyrophosphorylase small subunit 1, 2) (line 542), OsAGPL1 and OsAGPL2 (ADP-glucose pyrophosphorylase large subunit 1, 2) (line 543), RSR1 (rice starch regulator1) (line 554), OsNF-YB1 (nuclear transcription factor Y subunit B-1) (lines 555-556), GluB2 (glutelin precursor B2) (line 612), NF‐YC12 (nuclear transcription factor Y subunit C-12) (line 746), bHLH144 (basic helix-loop-helix protein 144) (lines 747-748), G‐box (CACGTG) (line 750), Wx (Waxy, granule-bound starch synthaseⅠ gene) (line 751) and CDS (coding sequence) (line 915) when these abbreviations appear in the text at the first time.

Reviewer 2 Report
This is an interesting article that detailed transcriptomics and metabolomics characterization of the OsMADS1Olr mutant, which was previously described as a mutant modifying grain shape in rice. As starch and seed storge proteins account for ~80% and ~10%, respectively, of dry weight of rice grains; the analysis of these traits in the genetic variability in race is important for rice breeding programs.
However, I have some serious concerns about the data provided on this manuscript; mainly the map-based cloning of the OsMADS1Olr mutant that was previously published in Li, P. et al 2020 (Characterization of the ‘Oat-Like Rice’ Caused by a Novel Allele OsMADS1Olr Reveals Vital Importance of OsMADS1 in Regulating Grain Shape in Oryza sativa L.. Rice 13, 73).
Which is the novelty presented on the current manuscript? Have the authors not observed previously that the short grain shape and the chalkiness in Olr endosperms were co-segregating?
Introduction
Some paragraphs are too long and bring information that it is still not expected at this point of the manuscript (even introducing figures and supplementary tables).
I refer mainly to paragraphs
Lines 58-75
Lines 80-96
Need to add more description of the previously paper (ref 22) which clearly described the phenotype of the OsMADS Old mutant to better understand why the authors want to go father on the characterisation of this mutant.
- Described that they are contradictory phenotypes between OsMADS1 mutants
- Describe phenotype of OsMADS Old mutant
Line 116 … relatively low filling rate and high expression of GBSSI under high temperature… (what GBSSI means?)
Line 129 we analyzed grain quality of the OsMADS1 Olr mutant and identified a correlation between the mutation in OsMADS1 gene and the chalkiness phenotype.
Line 148 generation at all tested locations in China (which are this location, where is these data?)
Line 152 remove between
Line 155 remove between
Line 157 rice(H,I)by (spaces)
Line 168 remove between
Line 171 To investigate (the) cause of the chalkiness
Line 189 starch and storage protein, (.) The starch and storage protein
Line 190. Compared with Nip, the percentage contents of total starch, amylose and amylopectin of Olr 191 brown rice are 86.66 ± 0.74%, 20.43 ± 0.15% and 66.23 ± 0.81% respectively, which is low- 192 ered by 10.51%, 4.55% and 12.35% respectively. (Please rephase this sentence)
Line 193 In contrast to the lower starch contents, 193 Olr brown rice accumulated extremely higher content of storage protein which is 14.13 ± 194 0.07% and increased by 51.77% compared with that of Nip (Table 1). (Please rephase this sentence)
Line 197 laso (also)
Line 255 and Table 1. We further speculated that both the poor grain quality and abnormal grain shape 225 of Olr were caused by the pleiotropic effects of the same mutated gene in it but not the 226 genetic background. It is difficult to understand this conclusion and the way that the analysis was done, have the authors first separated the grains based on their shape (long-Nip and short-Orl), and then they did the quality analysis?
Line 315 those of control plant. (you mean non transformed OsMADS1Olr mutant?)
Line 319 grain shape between T0 negative control plant (Control, Olr genetic background) (why the grain shape of the control OsMADS1Olr mutant is not the same than that shown in Fig 1A?, here you show an elongated rice)
Paragraphe 330-340 Here this is not clear. You introgressed wild-type OsMADS1 gene in Olr mutant and the phenotype (grain shape?, grain quality?) was or not complemented?
Line 350 RNAi T3 lines (SRi11, SRi14 and SRi16) showed a slightly darker brown color in the T4 (Have not the authors observe phenotypes in T0, T1, T2 lines? How to explain RNAi phenotypes in T3, T4?)
Could the authors provide a prove (qPCR) that MADS1 on those lines (T3-T4) is downregulated?
Line 375 dada (data)
Line 398-400 Sentence is redundant with line 401-403.
Line 425 Thirty (1-30) and thirty-six (1-36) robust clusters (what the 1- means?)
Line 441 We identified 30 and 30 storage protein related GO … (please rephase this sentence)
Chapter 2.9. Analysis of the Dynamic Gene Expression Patterns between Nip and Olr Grains (the all chapter is very descriptive and heave to read, maybe it is better to reduce it to the minimum and combine with 2.10.)
Concerning chapters 2.10 and 2.11, it is very descriptive too and an effort need to be done to resume the message, providing the most important indications of the genes/pathways that were different between both samples.
Concerning chapter 2.11, can be completely remove it and include a sentence indicating that similar results were obtained with qPCR.
Discussion
Several elements indicated in the discussion are extremely important to be change it to the introduction or the results to a better comprehension of the manuscript. It is a pity that authors omitted to describe the exact phenotype of exact phenotype of the Olr mutants (line 747-749), which is very important to do not be surprise with the phenotype shown in figure 4A.
The fact that other OsMADS1 mutants showing contradictory phenotypes with Olr can be mentioned in chapter 2.4. to understand why do you go farther in your analysis, as well as your complementation results.
Author Response
Responses to the comments of Reviewer 2
We highly appreciate your precious encouragement! Your comments are highly insightful and illuminating, and have enabled us to greatly improve the quality of our manuscript. We highly appreciate your kind help and valuable contributions to our manuscript! In the following pages are our point-by-point responses to your comments.
Reviewer 2
Comments and Suggestions for Authors
This is an interesting article that detailed transcriptomics and metabolomics characterization of the OsMADS1Olr mutant, which was previously described as a mutant modifying grain shape in rice. As starch and seed storge proteins account for ~80% and ~10%, respectively, of dry weight of rice grains; the analysis of these traits in the genetic variability in race is important for rice breeding programs.
Point 1: However, I have some serious concerns about the data provided on this manuscript; mainly the map-based cloning of the OsMADS1Olr mutant that was previously published in Li, P. et al 2020 (Characterization of the ‘Oat-Like Rice’ Caused by a Novel Allele OsMADS1Olr Reveals Vital Importance of OsMADS1 in Regulating Grain Shape in Oryza sativa L.. Rice 13, 73).
Which is the novelty presented on the current manuscript? Have the authors not observed previously that the short grain shape and the chalkiness in Olr endosperms were co-segregating?
Response: We highly appreciate these very important and reasonable questions raised by the wise reviewer.
Firstly, we admit that map-based cloning of the grain shape phenotype of OsMADS1Olr mutant was published in our previous paper. Secondly, we did observe that the grain quality of Olr (Oat-like rice) with relatively lower starch content but higher protein content is linked with its abnormal grain shape by the linkage analysis between the two traits. Thirdly, we also put forward the speculation that both the poor grain quality and abnormal grain shape of Olr were caused by the pleiotropic effects of the same mutated gene in it but not the genetic background. However, we were unable to completely confirm this speculation by these results. Thus, we decided to use another gene mapping and isolating method which is different from the map-based cloning method to confirm whether the poor grain quality of Olr was indeed and also caused by the pleiotropic effects of the same mutated gene which caused the abnormal grain shape of Olr. Accordingly, we adopted the mapping-by-sequencing method and finally confirmed our speculation by the mapping-by-sequencing result.
Introduction
Point 2-1: Some paragraphs are too long and bring information that it is still not expected at this point of the manuscript (even introducing figures and supplementary tables).
I refer mainly to paragraphs
Lines 58-75
Lines 80-96
Response: We appreciate the very important and valuable comments given by the reviewer.
We originally tried to introduce the background of gene expressions and regulatory networks of starch and storage protein metabolisms in rice grains by writing the two paragraphs and citing the figures and supplementary tables, which could be useful for the readers to understand the analysis and results of the corresponding Results 2.10. and 2.11..
After carefully learning the reviewer’s reasonable comments, we realized that the two paragraphs are indeed too long. Thus, we happily accepted the reviewer’s comments and have divided each of the two paragraphs into two shorter paragraphs, and deleted citations of the figures (Figure 6A, 7A). (lines 52-78, 79-103).
Point 2-2: Need to add more description of the previously paper (ref 22) which clearly described the phenotype of the OsMADS Old mutant to better understand why the authors want to go father on the characterisation of this mutant.
- Described that they are contradictory phenotypes between OsMADS1 mutants
- Describe phenotype of OsMADS Old mutant
Response: We thank the reviewer for giving us the wonderful suggestion.
(1) We have added more description of the previously paper (ref 22) and the detailed phenotype of the Olr mutant in the introduction part as “Thereinto, Olr is a spontaneous and severe OsMADS1 mutant, which was named for its unique grain shape which highly resembles oat grains. Olr displayed abnormal floral organs, open hulls formed by remarkably elongated leafy lemmas and paleae, occasionally formed conjugated twin brown rice, aberrant grain shape, low seed setting rate, slow grain-filling rate, low 1000-brown rice weight and extremely low yield [22].” (lines 141-145), and “The Olr mutant displays poor grain quality with chalky endosperms, abnormal morphology and loose arrangement of starch granules, lower starch content but higher protein content in grains.” (lines 162-164).
(2) We have described the contradictory phenotypes of grain quality between reported OsMADS1 mutants in the introduction part as “However, it’s interesting that the grain quality was only analyzed in NIL (SLG) [19], WYJ7-lgy3-dep1–1, RD23-lgy3-gs3, PA64S / 9311-lgy3-gs3 [20], which may be partly due to unavailable seeds caused by sterility or extremely low fertility of these mutants. These four rice NILs carry the same mutated allele of OsMADS1, OsMADS1lgy3 / OsLG3bSLG, but the effect of this allele to the grain quality of these NILs were divided into two distinct categories. There is no significant difference of chalkiness between grains of NIL (SLG) and its receptor parent Nip, which indicates that OsLG3bSLG did not affect grain quality in Nip [19]. However, WYJ7-lgy3-dep1–1, RD23-lgy3-gs3 and PA64S / 9311-lgy3-gs3 grains all displayed lower chalkiness compared with their respective receptor parent, WYJ7, RD23 and LYPJ (PA64S / 9311), which indicates that OsMADS1lgy3 affected and improved grain quality in WYJ7, RD23 and LYPJ [20]. Interestingly, the question that whether OsMADS1 regulates grain quality is still contradictory and obscure.” (lines 146-157).
Point 3: Line 116 … relatively low filling rate and high expression of GBSSI under high temperature… (what GBSSI means?)
Response: Thanks for the very important question and clue.
GBSSI means the granule-bound starch synthaseⅠ. To make it clear, we have revised the sentence to “In addition, Zhang et al. [14] reported that high temperature at the early filling stage greatly induced the expression of OsMADS7, and suppression of it in rice endosperm stabilized amylose content possibly by maintaining a relatively low filling rate and high expression of the encoding gene of GBSSⅠ (granule-bound starch synthaseⅠ) under high temperature stress.” (lines 121-126) from “In addition, Zhang et al. [14] reported that high temperature at the early filling stage greatly induced the expression of OsMADS7, and suppression of it in rice endosperm stabilized amylose content possibly by maintaining a relatively low filling rate and high expression of GBSSⅠ under high temperature stress.” (lines 113-116 in the original manuscript).
Point 4: Line 129 we analyzed grain quality of the OsMADS1 Olr mutant and identified a correlation between the mutation in OsMADS1 gene and the chalkiness phenotype.
Response: We appreciate the professional revision made by the reviewer.
We have happily accepted the reviewer’s revision and revised the content to “In this study, we analyzed grain quality of the OsMADS1Olr mutant (Olr) and identified a correlation between the mutation in OsMADS1 gene and the chalkiness phenotype.” (lines 160-162) from “In this study, we analyzed grain quality of the OsMADS1 mutant, Olr (Oat-like rice). We further found out that the poor grain quality of Olr was caused by the mutated OsMADS1Olr gene…….” (lines 129-133 in the original manuscript).
Point 5: Line 148 generation at all tested locations in China (which are this location, where is these data?)
Response: We thank the reviewer for pointing out the very important point.
The locations in which the Olr planted and observed are Sichuan and Hainan provinces of China. We investigated the phenotypes of Olr mutant by field observation in Sichuan and Hainan.
To make it clear, we revised the description to “Furthermore, the Olr displayed stable and heritable Olr phenotypes with chalky endosperms and abnormal grain shape in every generation in Sichuan and Hainan provinces of China by field observation, indicating the pleiotropic phenotypes are heritable and controlled by genetic factors.” (lines 181-184) from “Furthermore, the Olr displayed stable and heritable Olr phenotypes with chalky endosperms and abnormal grain shape in every generation at all tested locations in China, indicating the pleiotropic phenotypes are heritable and controlled by genetic factors.” (lines 146-149 in the original manuscript).
Point 6: Line 152 remove between
Response: Thanks for the correction.
We have removed the word “between” and corrected the content to “Appearance quality of Nip and Olr endosperms” (line 187) from “Appearance quality of endosperms between Nip and Olr” (line 152 in the original manuscript).
Point 7: Line 155 remove between
Response: Thanks for the correction.
We have removed the word “between” and corrected the content to “in the transverse sections of Nip (D-G) and Olr (H,K) brown rice” (line 190) from “in the transverse sections of brown rice between Nip (D-G) and Olr (H,K)” (line 155 in the original manuscript).
Point 8: Line 157 rice(H,I)by (spaces)
Response: Thanks for the correction.
We have added the “spaces” and corrected the content to “(D,E,H,I) Observation and analysis of chalkiness of Olr brown rice (H,I) by comparation with Nip (D,E) under dissecting microscope (D,H) and SEM (E,I).” (line 191) from “(D,E,H,I) Observation and analysis of chalkiness of Olr brown rice(H,I)by comparation with Nip (D,E) under dissecting microscope (D,H) and SEM (E,I).” (line 157 in the original manuscript).
Point 9: Line 168 remove between
Response: Thanks for the correction.
We have removed the word “between” and corrected the content to “Further cross-section analysis of Olr and Nip brown rice by dissecting microscope revealed that although both of their outer endosperms are translucent and similar to each other, the inner endosperm of Olr displayed a chalky white-core (Figure 1D,H)” (lines 203-205) from “Further cross-section analysis of brown rice between Olr and Nip by dissecting microscope revealed that although both of their outer endosperms are translucent and similar to each other, the inner endosperm of Olr displayed a chalky white-core (Figure 1D,H).” (lines 168-170 in the original manuscript).
Point 10: Line 171 To investigate (the) cause of the chalkiness
Response: Thanks for the correction.
We have added the word “the” and corrected the content to “To investigate the cause of the chalkiness in Olr endosperms, transverse sections of endosperms in Olr and Nip were further observed and analyzed by SEM (Scanning electron microscope).” (lines 205-207) from “To investigate cause of the chalkiness in Olr endosperms, transverse sections of endosperms in Olr and Nip were further observed and analyzed by SEM (Scanning electron microscope).” (lines 171-173 in the original manuscript).
Point 11: Line 189 starch and storage protein, (.) The starch and storage protein
Response: Thanks for the revision.
We have revised the sentence to “The abnormal starch granules and more chalkiness in Olr endosperms maybe further caused by the changes of relative contents of storage materials such as starch and storage protein. The starch and storage protein contents of brown rice from Olr, Nip and their progeny plants were subsequently determined to test this speculation.” (lines 221-224) from “Considering the abnormal starch granules and more chalkiness in Olr endosperms maybe further caused by the changes of relative contents of storage materials such as starch and storage protein, the starch and storage protein contents of brown rice from Olr, Nip and their progeny plants were subsequently determined to test this speculation.” (lines 187-190 in the original manuscript).
Point 12: Line 190. Compared with Nip, the percentage contents of total starch, amylose and amylopectin of Olr 191 brown rice are 86.66 ± 0.74%, 20.43 ± 0.15% and 66.23 ± 0.81% respectively, which is low- 192 ered by 10.51%, 4.55% and 12.35% respectively. (Please rephase this sentence)
Response: Thanks for the valuable suggestion.
We have rephase this sentence to “The percentages of total starch, amylose and amylopectin contents of Olr brown rice are 86.66 ± 0.74%, 20.43 ± 0.15% and 66.23 ± 0.81%, which are lowered by 10.51%, 4.55% and 12.35% than that of the Nip brown rice, respectively.” (lines 224-227).
Point 13: Line 193 In contrast to the lower starch contents, 193 Olr brown rice accumulated extremely higher content of storage protein which is 14.13 ± 194 0.07% and increased by 51.77% compared with that of Nip (Table 1). (Please rephase this sentence)
Response: Thanks for the valuable suggestion.
We have rephase this sentence to “By contrast, the Olr brown rice accumulated extremely higher content of storage protein, which is 14.13 ± 0.07% and increased by 51.77% than that of the Nip brown rice (Table 1).” (lines 227-229).
Point 14: Line 197 laso (also)
Response: Thanks for the correction.
We have corrected the word “laso” and the corresponding sentence to “These results that starch and storage protein contents were changed in Olr grains imply that metabolism and accumulation of starch and storage protein maybe also changed.” (lines 229-231) from “These results that starch and storage protein contents were changed in Olr grains imply that metabolism and accumulation of starch and storage protein maybe laso changed.” (lines 195-197 in the original manuscript).
Point 15: Line 255 and Table 1. We further speculated that both the poor grain quality and abnormal grain shape 225 of Olr were caused by the pleiotropic effects of the same mutated gene in it but not the 226 genetic background. It is difficult to understand this conclusion and the way that the analysis was done, have the authors first separated the grains based on their shape (long-Nip and short-Orl), and then they did the quality analysis?
Response: We appreciate the very important question raised by the reviewer.
Please let me make some explanations about the way that the analysis we done and the conclusion we made.
Firstly, a certain phenotype or trait is controlled by the genetic factors and affected by the environmental factors. We observed that the Olr mutant exhibited stable and heritable Olr phenotypes with chalky endosperms and abnormal grain shape in different environment (Sichuan and Hainan provinces). These results indicate that the Olr phenotype are heritable and controlled by the genetic factors.
Secondly, there is the possibility that both of the poor grain quality and the abnormal grain shape of Olr were caused by the pleiotropic effects of the same mutated gene in it; and there is also another possibility that the poor grain quality and the abnormal grain shape of Olr were caused by different genes.
Thirdly, we can test the above possibilities by the linkage analysis between the two traits of Olr mutant (the poor grain quality and the abnormal grain shape) by using a F2 or/and F3 segregating population. We constructed the F2 and F3 segregating populations by a cross between Olr (with the traits of poor grain quality and the abnormal grain shape) and Nip (Nipponbare) (with the traits of normal grain quality and normal grain shape). In the F2 and F3 segregating populations, if the poor grain quality and the abnormal grain shape are linked with each other, it indicates that the two traits are caused by the pleiotropic effects of the same mutated gene in the Olr mutant; otherwise, if the two traits s are segregated from each other, it indicates that the two traits are caused by different genes.
Fourthly, in the F2 and F3 segregating populations, we observed in the field that the grain shape of the plants in the F2 and F3 segregating populations can be divided into two apparently different groups. In the first group, these plants showed normal grain shape like Nip (F2 / F3 plants showing normal phenotypes); in the second group, these plants showed abnormal grain shape like Olr mutant (F2 / F3 plants showing Olr phenotypes). Subsequently, we separately mixed the seeds from differently individual F2 / F3 plants showing normal phenotypes, and separately mixed the seeds from differently individual F2 / F3 plants showing Olr phenotypes. And the quality analysis determining the starch and storage protein contents of the mixed samples from the F2 / F3 plants showing normal phenotypes, and the mixed samples from the F2 / F3 plants showing Olr phenotypes was performed, respectively.
Fifthly, our results show that the grain quality of Olr with relatively lower starch content but higher protein content is linked with its grain shape, which made us draw the speculation that “both the poor grain quality and abnormal grain shape of Olr were caused by the pleiotropic effects of the same mutated gene in it but not the genetic background”.
To make it clear and easier for the readers to understand, we have made the following revisions.
(1) We have revised the sentence to “To figure out whether changes of grain quality including starch and storage protein contents in Olr grains was linked with its abnormal grain shape, a linkage analysis between the two traits was performed. We measured the starch and storage protein contents of brown rice from plans of F2 and F3 segregating population derived from a cross combination between Olr and Nip.” (lines 241-245) from “To figure out whether changes of grain quality as well as starch and storage protein contents in Olr grains was linked with its abnormal grain shape, we measured the starch and storage protein contents of brown rice from plans of F2 and F3 segregating population derived from a cross combination between Olr and Nip.” (lines 208-211 in the original manuscript).
(2) We have added a sentence in the Results 2.2. to introduce the principle of the linkage analysis between the two traits we performed, which is “If the two traits are linked with each other, which indicates that they are caused by the same mutated gene in Olr; By contrast, the two traits are caused by different genes.” (lines 245-247).
(3) We have revised this speculation as “We further speculated that both the poor grain quality and abnormal grain shape of Olr were caused by the pleiotropic effects of the same mutated gene in it.” (lines 260-262) from “We further speculated that both the poor grain quality and abnormal grain shape of Olr were caused by the pleiotropic effects of the same mutated gene in it but not the genetic background.” (lines 225-227 in the original manuscript).
Point 16: Line 315 those of control plant. (you mean non transformed OsMADS1Olr mutant?)
Response: We thank the reviewer for raising this very important question.
The control plant means the T0 negative control plant which doesn’t harbor the exogenous wild-type OsMADS1 gene because the exogenous pUbi::OsMADS1 vector was failed to be transformed into the calli of OsMADS1Olr mutant (Olr) by Agrobacterium tumefaciens (EHA105)-mediated transformation.
To make it clear, we firstly revised the sentence to “Further statistical analysis showed that grain length, grain width and grain length-to-width ratio of these eight T0 positive plants were all significantly decreased than those of T0 negative control plant.” (lines 352-354) from “Further statistical analysis showed that grain length, grain width and grain length-to-width ratio of these eight T0 positive plants were all significantly decreased than those of control plant.” (lines 313-315 in the original manuscript).
To further explain and describe the T0 negative control plant, we secondly added a sentence in the caption of Fig. 4 as “The T0 negative control plant doesn’t harbor the exogenous wild-type OsMADS1 gene because the exogenous pUbi::OsMADS1 vector was failed to be transformed into the calli of Olr by Agrobacterium tumefaciens (EHA105)-mediated transformation.” (lines 360-362).
Point 17: Line 319 grain shape between T0 negative control plant (Control, Olr genetic background) (why the grain shape of the control OsMADS1Olr mutant is not the same than that shown in Fig 1A?, here you show an elongated rice)
Response: We thank the reviewer for raising this very reasonable question.
Please allow me to make some explanations. Firstly, the T0 negative control plant has the same grain shape with the OsMADS1Olr mutant (Olr). Secondly, the grains shown in Fig. 4A was composed of the outer lemmas and paleae as well as the inner brown rice. Thirdly, the endosperms in Fig. 1A were gained from the brown rice by removing the embryos and brans. The differences between brown rice and embryos of rice can be seen in Fig. 5A-H. Fourthly, compared with Nip, both OsMADS1Olr mutant (Li et al., 2000) and the T0 negative control plant has much longer grains (determined by the extremely longer lemmas and paleae) (Fig. 4A), but OsMADS1Olr mutant has smaller endosperms (Fig. 1A). Thus, the T0 negative control plant showed elongated grain shape in Fig. 4A, but the OsMADS1Olr mutant showed differently smaller endosperms.
To make it clear and easier for the readers to understand, we have revised description of the grain shape of OsMADS1Olr mutant in the Results 2.1 to “Our previous study revealed that Olr exhibits abnormal grain shape and long grains determined by elongated open lemmas and paleae, which is accompanied with lower 1000-brown rice weight and slower grain-filling rate [22].” (lines 173-175) from “Our previous study revealed that Olr exhibits abnormal grain shape accompanied with lower 1000-brown rice weight and slower grain-filling rate [22].” (lines 139-140 in the original manuscript).
Point 18: Paragraphe 330-340 Here this is not clear. You introgressed wild-type OsMADS1 gene in Olr mutant and the phenotype (grain shape?, grain quality?) was or not complemented?
Response: We thank the reviewer for pointing out this very important point.
- We originally tried to express the meanings in this paragraph as follows.
(1) The grain shape of Olr mutant was complemented by introducing the wild-type OsMADS1 gene, but we were unable to know whether the poor grain quality of Olr was complemented or not due to spikelet sterility or failure in germination of occasionally produced several seeds of the transgenic plants.
(2) And the spikelet sterility or low fertility of these T0 positive plants could be possibly attributed to the relatively cold weather during the flowering, grain filling and seed development stages of these plants.
- To make it clear and easier for the readers to understand, we have rephased and revised the corresponding paragraphs as the follows (lines 376-393).
These results showed that the grain shape of Olr was rescued by introducing the wild-type OsMADS1 gene, which confirmed that the abnormal grain shape of Olr was caused by its mutated OsMADS1Olr gene. However, we were unable to know whether the poor grain quality of Olr was recovered or not, due to spikelet sterility or failure in germination of occasionally produced several seeds of the transgenic plants. As shown in Fig. 4, both control plants and six of the eight T0 positive plants (C-28, C-30, C-3, C-4, C-5 and C-10) were sterile, only C-9 and C-6 plants occasionally produced several seeds but failed to germinate (Figure 4A, indicated by white arrows).
When the flowering and grain filling stages are in the summer seasons, the seed setting rate of Olr mutant is around 15% in Chengdu, Sichuan, China in different years. However, both T0 control and positive plants exhibited spikelet sterility or extremely low fertility (0.36% of C-9 and 1.05% of C-6 plants, respectively). So, we speculated that spikelet sterility or extremely low fertility of these T0 plants could be possibly attributed to the relatively cold weather in late October, early and middle November in Chengdu during the flowering, grain filling and seed development stages of these plants (Figure S1).
To study if OsMADS1 has a direct role in controlling grain quality , we further used a OsMADS1 seed-specific RNAi system and analyzed the grain appearance quality of descendant lines of pOsTip3::OsMADS1-RNAi plants in details.
Point 19: Line 350 RNAi T3 lines (SRi11, SRi14 and SRi16) showed a slightly darker brown color in the T4 (Have not the authors observe phenotypes in T0, T1, T2 lines (Point 19-1)? How to explain RNAi phenotypes in T3, T4? (Point 19-2))
Could the authors provide a prove (qPCR) that MADS1 on those lines (T3-T4) is downregulated? (Point 19-3)
Response: We thank the reviewer for raising the very important and illuminating questions.
(1) Response to Point 19-1: We initiatively observed the phenotypes of pOsTip3::OsMADS1-RNAi plants by field investigation in the T0 and T1 (The T1 lines are segregating populations) lines. Subsequently, we mainly analyzed phenotypes of the grain shape in the T2 lines.
(2) Response to Point 19-2: In this study, we focused on analyzing phenotypes in the homozygous and stable T3 lines. It should be noted that we actually analyzed grain appearance quality of the T4 seeds which were generated from the T3 plants (lines). We usually analyze phenotypes of transgenic rice like grain shape by using the homozygous T2 lines. In this study, we selected to analyze phenotypes of the homozygous and stable T3 lines mainly because we think that grain appearance quality of the T3 lines is more stable than the T2 lines which makes the corresponding statistical data is reliable.
In addition, to make it clear and easier for the readers to understand that we analyzed grain appearance quality of the T4 seeds generated from T3 plants, we revised the caption of Fig. 4 to” (A-H) Comparisons of grain chalkiness of brown rice (A-D) and endosperms (E-H) between homozygous T3 OsMADS1 seed-specific RNAi (pOsTip3::OsMADS1-RNAi) lines (SRi11, SRi14 and SRi16) and wild-type plants (Nip). T4 seeds generated from the corresponding T3 plants were analyzed.” (lines 404-407) from “(A-H) Comparisons of grain chalkiness of brown rice (A-D) and endosperms (E-H) between OsMADS1 seed-specific RNAi (pOsTip3::OsMADS1-RNAi) lines (SRi11, SRi14 and SRi16) and wild-type plants (Nip).” (lines 359-361 in the original manuscript).
(3) Response to Point 19-3: Although our previous analysis showed that expression of OsMADS1 gene could be suppressed by the pOsTip3::OsMADS1-RNAi system, we agree it’s a pity that we didn’t perform the expression analysis in the T3 OsMADS1 seed-specific RNAi lines in this manuscript. Nevertheless, we analyzed expression of the OsMADS1 gene in subsequent T4 OsMADS1 seed-specific RNAi lines (12 days after fertilization T5 seeds generated from the corresponding T4 line) by qRT-PCR analysis. The qRT-PCR result showed that expression of OsMADS1 gene in these lines were downregulated by about 80% compared with the negative control line (Figure 1 for responding to the reviewer). We didn’t add this result to the manuscript because this expression result was not from the T3 lines (Figure 5 in the main text), but from the subsequent T4 lines (Figure 1 for responding to the reviewer).
Figure 1 for responding to the reviewer. Relative expression of the OsMADS1 gene in T4 OsMADS1 seed-specific RNAi lines by qRT-PCR analysis. 12 DAF (days after fertilization) T5 seeds generated from the corresponding T4 OsMADS1 seed-specific RNAi (pOsTip3::OsMADS1-RNAi) lines (SRi-1, SRi-2 and SRi-3) were analyzed. The NC (negative control) line was used as a control. The value of OsActin mRNA was used as an internal control for data normalization, and the expression levels of OsMADS1 in NC were set as 1.0. Data presented are mean values ± SDs of three replicates. Student’s t-test: ***p < 0.001.
Point 20: Line 375 dada (data)
Response: Thanks for the correction.
We have corrected the word “dada” and the corresponding sentence to “Overview of mRNA-seq results indicate that the mRNA-seq data with sufficient and high-quality reads, high mapping ratios of these reads to the Nip reference genome, and good correlation and similarity within biological replicates can be used for the downstream gene expression detection and differentially expressed gene analysis (Table S3,4 and Figure S2).” (lines 419-423) from “Overview of mRNA-seq results indicate that the mRNA-seq dada with sufficient and high-quality reads, high mapping ratios of these reads to the Nip reference genome, and good correlation and similarity within biological replicates can be used for the downstream gene expression detection and differentially expressed gene analysis (Table S3,4 and Figure S2).” (lines 374-378 in the original manuscript).
Point 21: Line 398-400 Sentence is redundant with line 401-403.
Response: We thank the reviewer for pointing out the very important point.
We have deleted the redundant sentence “We initially performed GO enrichment analysis, which consists of three functional categories, namely CC (cellular components), BP (biological processes) as well as MP (molecular functions), and subsequently carried out KEGG pathway-based analysis.” (lines 401-403 in the original manuscript).
Point 22: Line 425 Thirty (1-30) and thirty-six (1-36) robust clusters (what the 1- means?)
Response: Thanks for the question pointed out by the scrupulous reviewer.
The “1” means the first robust cluster. To make it clear and easier for the readers to understand, we have revised the content to “Thirty and thirty-six robust clusters were identified in the robust clustering maps of Nip and Olr, respectively (Figure S5C,D).” (lines 464-465) from “Thirty (1-30) and thirty-six (1-36) robust clusters were identified in the robust clustering maps of Nip and Olr, respectively (Figure S5C,D).” (lines 425-426 in the original manuscript).
Point 23: Line 441 We identified 30 and 30 storage protein related GO … (please rephase this sentence)
Response: We thank the reviewer for giving the valuable suggestion.
We have rephased this sentence to “We identified 30 storage protein related GO items and KEGG pathways containing 376 genes which scattered in the 30 Nip robust clusters. Correspondingly, we also identified 30 storage protein related GO items and KEGG pathways containing 543 genes which scattered in the 36 Olr robust clusters.” (lines 474-477) from “We identified 30 and 30 storage protein related GO items & KEGG pathways containing 376 and 543 genes scattered in the 30 Nip and 36 Olr robust clusters, respectively.” (lines 441-443 in the original manuscript).
Point 24: Chapter 2.9. Analysis of the Dynamic Gene Expression Patterns between Nip and Olr Grains (the all chapter is very descriptive and heave to read, maybe it is better to reduce it to the minimum and combine with 2.10.)
Response: We thank the reviewer for the valuable and wonderful suggestion.
We agree that this chapter is very descriptive. Thus, we happily accepted the reviewer’s suggestion, and have tried our best to reduce the contents of this chapter to the minimum (lines 457-500). About 35% content of this chapter (from 903 words to 582 words which contains the most important information of Figure S5, 6 and Datasets S2-7) has been cut down. The contents in the lines 420-421, 423-425, 427-429, 431-435, 438, 444-447, 449-451, 455-460, 455-460, 462-464 and 466-472 of the original manuscript have been deleted or cut down. Because the content of this chapter is relatively independent with the chapter 2.10, we suggest to keep it as an independent chapter. Of course, if the reviewer further thinks it is better to combine this chapter with chapter 2.10., we will follow the suggestion.
Point 25: Concerning chapters 2.10 and 2.11, it is very descriptive too and an effort need to be done to resume the message, providing the most important indications of the genes/pathways that were different between both samples.
1175 882
Response: We thank the reviewer for the valuable and wonderful suggestion.
We agree that this chapter is also descriptive. Thus, to provide the most important information of the gene expressions and regulatory networks different between Olr and Nip, we have tried our best to resume the message and thoroughly revise the two chapters (2.10 and 2.11) as follows (lines 501-575, 576-635).
(1) The contents in the lines 519-520, 523-524, 526-532, 519-520, 534-536, 538-542, 545-548, 551, 553-556, 557-560, 562-564 and 567-569 of chapter 2.10 in the original manuscript have been deleted, cut down or revised.
(2) The contents in the lines 581-583, 612-614, 616-617, 619-620, 621-629, 632-635 and 632-640 of chapter 2.11 in the original manuscript have been deleted, cut down or revised.
Point 26: Concerning chapter 2.12, can be completely remove it and include a sentence indicating that similar results were obtained with qPCR.
Response: We thank the reviewer for giving the valuable suggestion.
We have removed the chapter 2.12 and added a sentence “Furthermore, expression patterns of OsMADS1 and other 12 DEGs involved in seed storage protein biosynthesis and regulation in Olr and Nip grains were very similar between qRT-PCR results and mRNA-seq data, which validated the mRNA-seq results (Figure 8; Figure S7).” in the end of chapter 2.11. (lines 635-639).
Discussion
Point 27-1: Several elements indicated in the discussion are extremely important to be change it to the introduction or the results to a better comprehension of the manuscript.
Response: We thank the reviewer for giving us the wonderful and illuminating suggestion.
We have moved the introduction of the contradictory phenotypes in grain quality among previously reported OsMADS1 mutants from the first paragraph of Discussion part (lines 681-696 in the original manuscript) to the Introduction part (lines 135-169) as the follows.
And after the first OsMADS1 mutant, lhs1 (leafy hull sterile1) was reported by Kinoshita et al. [23] at least 11 mutants and four NILs (Near-isogenic lines) of OsMADS1 including lhs1 [15,23], nsr (naked seed rice) [17], NF1019, ND2920, NE3043 and NG778 [16], osmads1-z [18], afo (abnormal floral organs) [24], ohms1 (open hull and male sterile 1) [25], cy15 [26] and Olr (Oat-like rice) [22], NIL (SLG) [19], WYJ7 (Wuyungeng No. 7)-lgy3-dep1–1, RD23-lgy3-gs3 and PA64S (Peiai 64S) / 9311 (Yangdao No. 6)-lgy3-gs3 [20]have been reported so far. Thereinto, Olr is a spontaneous and severe OsMADS1 mutant, which was named for its unique grain shape which highly resembles oat grains. Olr displayed abnormal floral organs, open hulls formed by remarkably elongated leafy lemmas and paleae, occasionally formed conjugated twin brown rice, aberrant grain shape, low seed setting rate, slow grain-filling rate, low 1000-brown rice weight and extremely low yield [22].
However, it’s interesting that the grain quality was only analyzed in NIL (SLG) [19], WYJ7-lgy3-dep1–1, RD23-lgy3-gs3, PA64S / 9311-lgy3-gs3 [20], which may be partly due to unavailable seeds caused by sterility or extremely low fertility of these mutants. These four rice NILs carry the same mutated allele of OsMADS1, OsMADS1lgy3 / OsLG3bSLG, but the effect of this allele to the grain quality of these NILs were divided into two distinct categories. There is no significant difference of chalkiness between grains of NIL (SLG) and its receptor parent Nip, which indicates that OsLG3bSLG did not affect grain quality in Nip [19]. However, WYJ7-lgy3-dep1–1, RD23-lgy3-gs3 and PA64S / 9311-lgy3-gs3 grains all displayed lower chalkiness compared with their respective receptor parent, WYJ7, RD23 and LYPJ (Liangyoupeijiu, PA64S / 9311), which indicates that OsMADS1lgy3 affected and improved grain quality in WYJ7, RD23 and LYPJ [20]. Interestingly, the question that whether OsMADS1 regulates grain quality is still contradictory and obscure.
Point 27-2: It is a pity that authors omitted to describe the exact phenotype of exact phenotype of the Olr mutants (line 747-749), which is very important to do not be surprise with the phenotype shown in figure 4A.
Response: We thank the reviewer for the valuable comment.
We have increased the description of the exact phenotype of the Olr mutant to “On the other hand, the Olr mutant harboring OsMADS1Olr exhibited decreased grain quality and abnormal grain shape. Olr displayed chalky endosperms with opaque white-core in the inner endosperm, abnormal morphology and loose arrangement of starch granules, lower starch content but higher protein content in grains. In addition, Olr showed abnormal and long grains, open hull formed by remarkably elongated leafy lemmas and paleae, extremely low seed setting rate, slow grain-filling rate, small brown rice, occasionally formed conjugated twin brown rice, low 1000-brown rice weight and extremely low yield.” (lines 706-713) from “On the other hand, the Olr mutant harboring OsMADS1Olr exhibited decreased grain quality accompanying with extremely low spikelet fertility, open hull formed by remarkably elongated leafy lemmas and paleae, and extremely low yield.” (lines 747-750 in the original manuscript).
Point 28: The fact that other OsMADS1 mutants showing contradictory phenotypes with Olr can be mentioned in chapter 2.4. to understand why do you go farther in your analysis, as well as your complementation results.
Response: We thank the reviewer for the valuable suggestion.
We have mentioned other OsMADS1 mutants showing contradictory phenotypes with Olr in the chapter 2.4. by adding two sentences as “Liu et al. [20] reported that three NILs of the mutated OsMADS1lgy3 gene, WYJ7-lgy3-dep1–1, RD23-lgy3-gs3 and PA64S / 9311-lgy3-gs3, all displayed improved grain quality. This result is contradictory with the poor grain quality of Olr.” (lines 265-267).
